# Chimeric systems composed of swapped Tra subunits between distantly-related F plasmids reveal striking plasticity among type IV secretion machines

**Kouhei Kishida**[¤a], **Yang Grace Li**, **Natsumi Ogawa-Kishida**[¤b], **Pratick Khara, Abu Amar M. Al Mamun, Rachel E. Bosserman**[¤c]**, Peter J. Christie**\*

Department of Microbiology and Molecular Genetics, McGovern Medical School at UTHealth, Houston, Texas, United States of America

¤a  Current address: Kouhei Kishida, Graduate School of Life Sciences, Tohoku University, Katahira, Aobaku, Sendai, Japan
¤b  Current address: Natsumi Ogawa-Kishida, Graduate School of Life Sciences, Tohoku University, Katahira, Aobaku, Sendai, Japan
¤c  Current address: Rachel Bosserman. Department of Pathology and Immunology, Division of Laboratory and Genomic Medicine, Washington School of Medicine, St. Louis, Missouri, United States of America
* Peter.J.Christie@uth.tmc.edu

**Data Availability Statement:** All data are fully available without restriction. All relevant data are

## Abstract

Bacterial type IV secretion systems (T4SSs) are a versatile family of macromolecular translocators, collectively able to recruit diverse DNA and protein substrates and deliver them to a wide range of cell types. Presently, there is little understanding of how T4SSs recognize substrate repertoires and form productive contacts with specific target cells. Although T4SSs are composed of a number of conserved subunits and adopt certain conserved structural features, they also display considerable compositional and structural diversity. Here, we explored the structural bases underlying the functional versatility of T4SSs through systematic deletion and subunit swapping between two conjugation systems encoded by the distantly-related IncF plasmids, pED208 and F. We identified several regions of intrinsic flexibility among the encoded T4SSs, as evidenced by partial or complete functionality of chimeric machines. Swapping of VirD4-like TraD type IV coupling proteins (T4CPs) yielded functional chimeras, indicative of relaxed specificity at the substrate—TraD and TraD—T4SS interfaces. Through mutational analyses, we further delineated domains of the TraD T4CPs contributing to recruitment of cognate vs heterologous DNA substrates. Remarkably, swaps of components comprising the outer membrane core complexes, a few F-specific subunits, or the TraA pilins supported DNA transfer in the absence of detectable pilus production. Among sequenced enterobacterial species in the NCBI database, we identified many strains that harbor two or more F-like plasmids and many F plasmids lacking one or more T4SS components required for self-transfer. We confirmed that host cells carrying co-resident, non-selftransmissible variants of pED208 and F elaborate chimeric T4SSs, as evidenced by transmission of both plasmids. We propose that T4SS plasticity enables the facile assembly of functional chimeras, and this intrinsic flexibility at the structural level can

within the paper and its Supporting information files.

**Funding:** This work was supported by the National Institute of General Medical Sciences of the National Institutes of Health (R01GM48746 and R35GM131892 to PJC) and the National Institute of Allergy and Infectious Diseases of the National Institutes of Health (R21AI159970 to PJC). The funders played no role in study design, data collection and analysis, decision to publish, or preparation of the manuscript.

**Competing interests:** The authors have declared that no competing interests exist.

account for functional diversification of this superfamily over evolutionary time and, on a more immediate time-scale, to proliferation of transfer-defective MGEs in nature.

## Author summary

Mobile genetic elements (MGEs) comprise a diverse group of extrachromosomal plasmids or integrated DNA fragments that are widely distributed among many bacterial species. MGEs typically encode conjugation systems dedicated to their transmission to other bacteria, and also code for resistance to antibiotics or virulence or other fitness traits. The conjugation systems, along with an equally medically important group of translocators devoted to the interkingdom delivery of protein effectors by pathogenic species, comprise the superfamily of type IV secretion systems (T4SSs). Recent studies have defined many mechanistic and structural features of the T4SSs, yet there remains little understanding of how T4SSs recruit specific DNA or protein substrates, elaborate functional channels, and in some cases build attachment organelles termed conjugative pili. We explored the mechanics of T4SS machine function by systematically exchanging individual components between two distinct conjugation systems functioning in enterobacterial species. Through construction of chimeric machines, and further mutational analyses, we identified subunits or protein domains of conjugation machines specifying recruitment of different DNA substrates or selectively contributing to assembly of translocation channels or conjugative pili. Such features of T4SSs are prime targets for development of inhibitory strategies aimed at blocking T4SS functions for therapeutic intervention.

## Introduction

The bacterial type IV secretion systems (T4SSs) comprise a functionally and structurally diverse superfamily of macromolecular translocators [1–3]. In Gram-negative bacteria, most T4SSs consist of three major substructures, the substrate receptor or type IV coupling protein (T4CP), an inner membrane complex (IMC), and an outer membrane core complex (OMCC) [4, 5]. A major subfamily of the T4SSs, the conjugation machines functioning in Gram-negative bacteria, also elaborate conjugative pili [6]. In general, the core structural features of T4SSs are built from a conserved set of subunits generically termed the VirD4 T4CP and VirB1—VirB11 [7]. At this time, high-resolution structures have been presented for several T4SS subunits, including the VirD4 T4CP and the VirB4 and VirB11 ATPases that energize substrate recruitment and trafficking [5,8,9]. Additionally, structures have been solved of OMCCs from several model T4SSs [10–15] and a number of conjugative pili [16–19]. Most recently, a nearly intact T4SS encoded by conjugative plasmid R388 (T4SS$_{R388}$) was solved at near-atomic resolution [5].

These *in vitro* structures have supplied important information about the architectures of several model T4SSs in their quiescent states. The structures, coupled with other biochemical and genetic findings, also predict regions of conformational flexibility that might be important for structural transitions associated with machine activation. These inherently flexible regions include the T4CP—channel interface, as suggested by difficulties in isolation of stable T4CP—channel complexes and recent evidence for spatial repositioning of VirD4-like TrwB to polar regions of donor cells upon contact with recipient cells, which is where T4SS$_{R388}$ channels are located [20]. In the T4SS$_{R388}$ structure, the IMC presents as an intrinsically stable substructure with well-defined contacts among its VirB3-, VirB4-, VirB5-, VirB6-, and VirB8-like

components [5]. However, there is also evidence for different conformations of T4SS IMCs and of VirB4 ATPases, possibly reflecting different functional states [5,21,22]. The $T4SS_{R388}$ possesses a central periplasmic stalk composed of pentamers of VirB6 with VirB5 with well-defined contacts [5], but the stalk lacks a central channel which presumably must form for substrate transfer or pilus production. Finally, the VirB7-, VirB9-, and VirB10-like components of the $OMCC_{R388}$ assemble as well-defined inner and outer (I and O) layers, but these substructures are only sparsely connected to each other and adopt distinct rotational symmetries. Such features are also prominent among OMCCs of other T4SSs solved to date, leading to proposals that the OMCCs are highly structurally dynamic [2,12–15].

Recently, we reported the structure of the OMCC encoded by the IncFV plasmid pED208 (hereafter $OMCC_{ED}$) [15]. The $OMCC_{ED}$ has a central cone (CC; equivalent to the I-layer of $T4SS_{R388}$) with 17-fold symmetry and an outer ring complex (ORC, equivalent to the R388 O-layer) with 13-fold symmetry. The CC is composed of 17 copies of the C-terminal β-barrel domain of VirB10-like TraB and between 13 and 17 copies of the N-terminal domain (NTD) of VirB7-like TraV. The ORC is composed of 26 copies each of VirB9-like TraK and C-terminal domains (CTDs) of TraV. The CC and ORC are connected only by linkers connecting NTDs and CTDs of TraV, and short stretches of TraB that interact with TraK underneath the β-barrel domains. In view of these sparse connections, we proposed that the CC and ORC substructures are capable of independent movement. Remarkably, results of structure-guided mutational analyses established, first, that deletion of the entire TraV linker domain is completely dispensable for function of the pED208-encoded T4SS ($T4SS_{ED}$). Second, deletion of TraK from the $T4SS_{ED}$ does not abolish substrate transfer, but does block production of the pED208-encoded (ED208) pilus [15]. Overall, these findings suggest that the $T4SS_{ED}$ accommodates major structural changes without effects on substrate transfer, although some mutations can selectively block piliation.

In this study, we further interrogated the plasticity of F-encoded T4SSs by assessing the functionality of chimeric machines. The chimeras were generated by swapping individual Tra/Trb subunits between T4SSs encoded by the phylogenetically distantly-related F-like plasmids, classical F and pED208 [23–25]. F is designated as $IncF1B/MOB_{F12}$ group A and pED208 and its progenitor plasmid $F_0$-*lac* are $IncFV/MOB_{F12}$ group C plasmids [26]. In line with their distinct Inc/MOB classifications, the Tra/Trb homologs exhibit considerable sequence divergence (**S1A Fig**). F and ED208 pili also differ serologically and render host cells susceptible to different male-specific bacteriophages [27, 28]. We systematically deleted each of the *tra/trb* genes from F and pED208, and then tested whether subunit swaps between these systems yielded functional chimeric T4SSs. We identified chimeric machines with swaps or mutations of specific machine components that conferred altered DNA substrate selection or the capacity to function as DNA transfer channels in the absence of detectable pilus production. We also determined that host cells with co-resident pED208 and F plasmids deleted of essential *tra/trb* genes naturally exchange subunits to build functional chimeric machines. We propose that the intrinsic capacity of F systems, and possibly other T4SSs, to incorporate heterologous machine components not only expands and diversifies machine functions, but also serves as a mechanism for proliferation of non-selftransmissible MGEs in nature.

## Results and discussion

### Diversification of the substrate repertoire through plasticity at the T4CP—T4SS interface

**TraD swapping supports transfer of noncognate substrates.** To begin interrogating the crossfunctionality of the pED208 and F systems, we first asked whether the DNA transfer and

replication (Dtr) proteins involved in processing these plasmids for transfer recognize the non-cognate substrates. Conjugative DNA transfer initiates by assembly of Dtr proteins at the MGE's origin-of-transfer (*oriT*) sequence, forming the relaxosome. The catalytic subunit of the relaxosome, the relaxase, nicks the DNA strand destined for transfer (T-strand) within the *oriT* sequence and remains covalently attached to the 5' end of the T-strand [29]. Concomitantly, the VirD4-like T4CP recruits the DNA substrate through recognition of translocation signals (TSs) carried by the relaxase and other Dtr proteins [30,31]. To determine if the pED208-encoded Dtr factors recognize F's *oriT* sequence and vice versa, we constructed p*oriT* plasmids carrying the *oriT* sequences of pED208 or F. Host cells harboring pED208 and p*oriT*$_{ED}$ transferred both plasmids at equivalent frequencies, and similarly for host cells harboring F and p*oriT*$_F$ (**S2A Fig**). However, hosts with pED208 and p*oriT*$_F$ failed to transfer p*oriT*$_F$, and similarly for hosts with F and p*oriT*$_{ED}$. These findings, coupled with results of T4CP swapping studies presented below, strongly indicate that the pED208- and F-encoded Dtr factors fail to bind or process the noncognate p*oriT* substrates for transfer.

In F-like systems, the VirD4-like TraD receptors couple F plasmids to their cognate T4SS channels [32]. To test whether F-encoded TraD (TraD$_F$) recognizes the pED208 substrate, we constructed donors carrying pED208Δ*traD* and producing TraD$_F$ from a separate plasmid (**Fig 1A**). Remarkably, TraD$_F$ supported transfer of pED208Δ*traD* through the pED208-encoded T4SS (T4SS$_{ED}$) at frequencies only ~$10^2$-fold lower than observed for TraD$_{ED}$-mediated transfer of wild-type (WT) pED208 (**Fig 1Bi**). Hosts with FΔ*traD* and producing TraD$_{ED}$ from a separate plasmid also transferred FΔ*traD* through the F-encoded T4SS (T4SS$_F$), albeit at lower frequencies (~5 x $10^{-7}$ transconjugants/donor, Tcs/D) than observed for TraD$_F$-mediated F transfer (**Fig 1Ci**). Both TraD T4CPs thus functionally interacted with the noncognate plasmid substrates and T4SSs, although TraD$_F$ coupled the pED208 substrate to the T4SS$_{ED}$ channel considerably more efficiently than TraD$_{ED}$ functioned in the F system. Both TraD proteins produced from separate plasmids, as well as TraD deletion mutants and chimeric proteins described below, accumulated at detectable levels as shown by immunostaining of N-terminal Strep-tagged variants; we also confirmed that the Strep-tag did not affect TraD functionality (**S2B and S2C Fig**).

**Distinct regions of TraD contribute to recruitment of cognate and noncognate substrates.** F-like TraD subunits consist of an N-terminal transmembrane domain (NTD) implicated in establishing contacts with IMC components, a central nucleotide binding domain (NBD) thought to provide the energy for early-stage substrate processing and transfer reactions, and an unstructured C-terminal domain (CTD) involved in substrate binding (**Fig 1A**) [4,32,33]. The NBDs of TraD$_{ED}$ and TraD$_F$ have conserved primary sequences and adopt similar structures as predicted by Alphafold [34], whereas their NTDs and CTDs are more divergent (**S1B Fig**). Previous studies have determined that the C-terminal ~8 residues of TraD$_F$ specifically interacts with TraM$_F$, a Dtr factor that binds the *oriT*$_F$ sequence and promotes formation of the catalytically-active relaxosome through recruitment of the TraI$_F$ relaxase [35,36]. The TraD$_F$—TraM$_F$ contact thus physically couples the F plasmid substrate to the T4SS$_F$.

Here, we determined that deletion of 15 C-terminal residues (ΔC15) from TraD$_F$ and TraD$_{ED}$ attenuated transfer respectively of F and pED208 through their cognate T4SSs (**Fig 1Bii and 1Cii**), likely due to loss of the TraD—TraM interactions. We also swapped the C-terminal 15 residues (C15) between TraD$_F$ and TraD$_{ED}$, and found that the TraD$_{ED}$C15$_F$ chimera supported transfer of pED208 through the T4SS$_{ED}$ at a frequency comparable to that conferred by the TraDΔC15$_{ED}$ truncation (**Fig 1Bii**). Similarly, TraD$_F$C15$_{ED}$ supported transfer of F through the T4SS$_F$ at a frequency similar to that conferred by the TraDΔC15$_F$ truncation (**Fig 1Cii**). In line with previous findings [35–37], the C15 motifs of TraD$_{ED}$ and TraD$_F$ thus function specifically in recruitment of their cognate pED208 and F plasmid substrates.

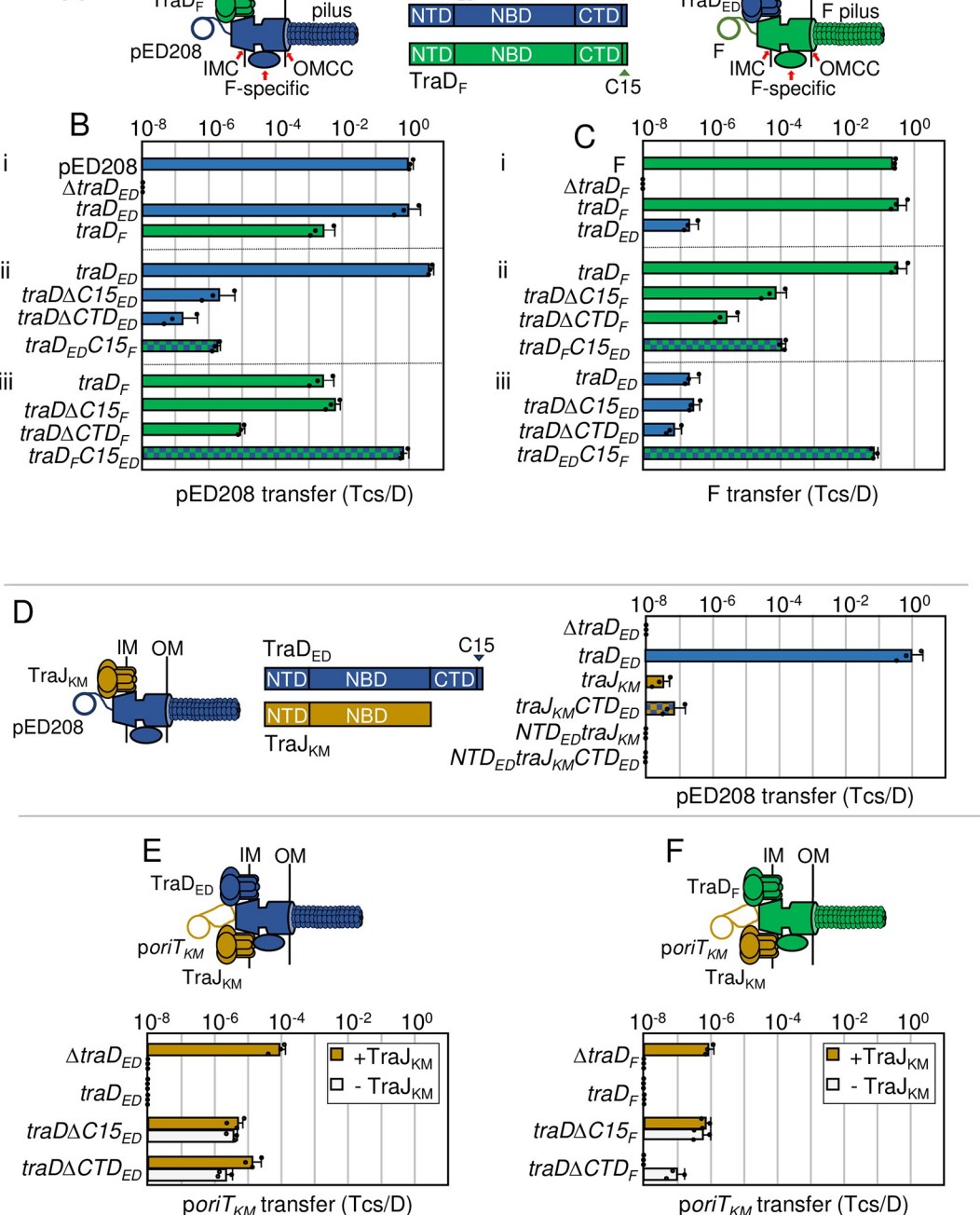

**Fig 1. Chimeric F-like T4SSs with substituted T4CPs retain function. A)** Schematics depicting chimeric T4SSs with substituted TraD subunits. Left: Chimera of pED208-encoded T4SS (components in blue) with F-encoded TraD (green). Right: F-encoded T4SS (green) with pED208-encoded TraD (blue). IM, inner membrane; OM, outer membrane. IMC, inner membrane complex; F-specific, F-specific complex(es); OMCC, outer membrane core complex. Center: Schematic of TraD T4CPS with domains shown as N-terminal transmembrane domain (NTD), nucleotide binding domain (NBD), C-terminal domain (CTD), C-terminal discrimination motif (C15). **B & C)** pED208 and F transfer frequencies presented as transconjugants per donor (Tcs/D). **i)** Donors carrying pED208 or F and the corresponding $\Delta traD$ mutant plasmids without or with plasmids expressing $traD_{ED}$ or $traD_F$. **ii)** Donors carrying the pED208 or F $\Delta traD$ mutant plasmids additionally expressing cognate $traD$ genes, truncations shown, or $traD$ chimeras with swapped C15 discrimination motifs. **iii)** Donors carrying the $\Delta traD$ mutant plasmids additionally expressing the noncognate $traD$ genes, truncations shown, or $traD$ chimeras with swapped C15 discrimination motifs. For this and other figures/panels in this manuscript, blue and green bars respectively denote the pED208 and F plasmids or plasmid sources of the genes or truncations, and blue/green checkered bars denote a chimeric gene. **D)** Left: Schematic of chimeric T4SS$_{ED}$ (blue) with substituted TraJ T4CP from pKM101

(yellow). Middle: Schematic of TraD$_{ED}$ and TraJ$_{KM}$ with domains shown. Right: Transfer frequencies by donors carrying pED208Δ*traD* donors and plasmids expressing *traD$_{ED}$* (blue bars) *traJ$_{KM}$* (yellow bars), or *traJ$_{KM}$* chimeric proteins with fused TraD$_{ED}$ domains (blue/yellow checkered bars) shown. **E & F)** Upper: Schematics of T4SSs with TraD or substituted TraJ$_{KM}$ translocating the p*oriT$_{KM}$* substrate. Lower: p*oriT$_{KM}$* transfer frequencies by donors carrying the pED208 and F Δ*traD* variants shown and plasmids expressing the intact or truncated forms of *traD* shown. In these experiments, color coding represents p*oriT* transfer in the presence (yellow bars) or absence (white bars) of TraJ$_{KM}$. All matings were repeated at least three times in triplicate; a representative experiment is shown with replicate data points and the average transfer frequencies as horizontal bars along with standard deviations as error bars.

Importantly, even in the absence of their C15 motifs, both TraD truncations supported transfer of their cognate substrates, suggesting that other domains of these T4CPs also contribute to substrate binding and recruitment. Consistent with this proposal, deletions of the larger, nonconserved CTDs (166-residues for TraD$_F$, 148 residues for TraD$_F$; **S1B Fig**) attenuated transfer of both plasmids through their respective T4SSs by ~$10^{1-1.5}$-fold relative to effects of the ΔC15 mutations (**Fig 1Bii** and **1Cii**). Moreover, even the ΔCTD truncation mutants (designated NTD/NBDs) supported transfer of the cognate plasmids at frequencies well above threshold levels of detection ($<10^{-8}$ Tcs/D) (**Fig 1Bii** and **1Cii**), suggesting that contacts likely mediated by the NBDs contribute to substrate docking. Overall, the C15 motifs conferred $10^{4}$$^{-5}$-fold stimulatory effects on transfer of the cognate substrates, the CTDs devoid of the C15 motifs conferred ~$10^{1-1.5}$-fold stimulatory effects, and the NTD/NBD regions devoid of the CTDs conferred ~$10^{1-2.5}$-fold stimulatory effects.

Having shown that TraD$_F$ integrates into the pED208 system (**Fig 1Bi**), we sought to identify the TraD$_F$ domains contributing to recognition of the pED208 substrate. As expected, deletion of the C15 discrimination motif had no effect on TraD$_F$ -mediated transfer of pED208 through the T4SS$_{ED}$ (**Fig 1Biii**). Further analyses of truncation mutants indicated that TraD$_F$'s CTD and NTD/NBD regions each conferred ~$10^{3}$-fold stimulatory effects on pED208 transfer (**Fig 1Biii**). Although TraD$_{ED}$ supported only a low level of F transfer through the F system, the corresponding analyses of TraD$_{ED}$ truncation mutants in the F system showed the same general pattern: TraD$_{ED}$'s C15 motif did not contribute to F transfer and the CTD and NTD/NBD regions each stimulated transfer by ~$10^{0.5-1}$-fold (**Fig 1Ciii**). Sequence or structural motifs within TraD's CTD and likely the NBD thus also contribute to recruitment of noncognate substrates. Finally, in the pED208 system, the TraD$_F$C15$_{ED}$ chimera elevated transfer of pED208 to levels similar to that conferred by native TraD$_{ED}$, and the TraD$_{ED}$C15$_F$ chimera also efficiently conveyed F through the T4SS$_F$ (**Fig 1Biii** and **1Ciii**). These findings further establish the discriminatory role of the TraD C15 motifs for recruitment of cognate F-like plasmids.

We next asked whether a T4CP that naturally lacks a CTD also functionally integrates into the F and pED208 systems. The TraJ T4CP is required for transfer of the IncN plasmid pKM101 through the pKM101-encoded T4SS (T4SS$_{KM}$) [38]; it is only weakly related to TraD$_{ED}$ throughout its length, although the predicted NBD fold resembles that of TraD$_{ED}$ (**S1B Fig**). Interestingly, donors harboring pED208Δ*traD* and producing TraJ$_{KM}$ supported transfer of pED208Δ*traD*, albeit at a low frequency of ~$10^{-7}$ Tcs/D (**Fig 1D**). We appended TraD$_{ED}$'s CTD to TraJ$_{KM}$, which did not significantly impact pED208 transfer. We also substituted TraD$_{ED}$'s NTD for that of TraJ$_{KM}$, but this completely abolished pED208 transfer (**Fig 1D**), possibly reflecting the importance of functional or structural interactions between the N-terminal transmembrane and nucleotide-binding domains of T4CPs [39–41]. TraJ$_{KM}$ mediates the transfer of p*oriT$_{KM}$*, a plasmid harboring the pKM101 *oriT* sequence, through the T4SS$_{KM}$ at a frequency of ~$10^{0}$ Tcs/D on solid surfaces [38]. In the absence of TraD, TraJ$_{KM}$ also supported transfer of p*oriT$_{KM}$* through the T4SS$_{ED}$ and T4SS$_F$ channels at moderate frequencies of $10^{-4}$–$10^{-6}$ Tcs/D (**Fig 1E and 1F**). TraJ$_{KM}$ thus functionally interacts with both F-like T4SSs,

although not as efficiently as with the cognate $T4SS_{KM}$ system. We next tested whether $TraD_{ED}$ and $TraD_F$ recruit the $poriT_{KM}$ substrate. Although full-length versions of these T4CPs failed to support $poriT_{KM}$ transfer, the ΔC15 or ΔCTD truncation mutants supported transfer (**Fig 1E and 1F**). TraD's NTD/NBD thus recruits a completely heterologous substrate, whereas the C15 motif apparently blocks this recruitment. Finally, strains co-producing native $TraD_{ED}$ or $TraD_F$ and $TraJ_{KM}$ failed to transfer $poriT_{KM}$, suggesting that the TraD T4CPs outcompete $TraJ_{KM}$ for engagement with the cognate T4SS channels (**Fig 1E and 1F**).

**Summary.** With the sole exception of the structurally-defined TraD CT—TraM interaction [35], the nature of T4CP—relaxosome contacts is poorly understood for any conjugation system. Here, we supplied further genetic evidence that TraD's CT discrimination motif plays an important role in recruitment of cognate F or pED208 substrates, and also actively blocks other TraD motifs from productively engaging with the heterologous pKM101 substrate. Outside of the CT motif, TraD carries sequence or structural motifs in its CTD and likely NBD that contribute to recruitment of cognate F-like plasmids as well as noncognate DNA substrates. These findings are compatible with evidence that other T4CPs that lack C-terminal extensions are capable of productive engagement with cognate DNA substrates, in some cases through binding of relaxosomal components other than relaxases [38, 42]. Relaxases such as $TraI_F$ have been shown to harbor internal translocation signals (TSs), while other relaxases can carry C-terminal TSs that are required for DNA substrate docking and transfer [22,31,43]. We envision that TraD and other T4CPs have evolved to carry distinct motifs in their NBDs and, when present, C-terminal extensions that are capable of binding different TSs harbored by relaxosome components. In some cases, as with the TraD CT—TraM interaction, these might be highly specific contacts that confer recognition of only one substrate. In other cases, as with motifs carried by TraD's NBD and CTD, more promiscuous engagement of various heterologous DNA substrates might be afforded through recognition of conserved sequence or structural features shared by all relaxosomes, e.g., by virtue of a common DNA enzymology. Regardless of the underlying mechanism(s), the relaxed specificity of T4CPs for recruitment of heterologous substrates even at low frequencies sets the stage for acquisition of mutations enabling elevated transfer as well as further substrate diversification.

## IMC components are not swappable

The pED208- and F-encoded IMCs are composed of VirB3-like TraL, VirB4-like TraC, VirB6-like TraG, and VirB8-like TraE [44–46]. The homologs exhibit between 24 to 55% sequence identities, but are predicted to adopt similar structural folds (**S1A and S1C Fig**). Deletion of IMC genes from both F-like plasmids abolished DNA transfer, and complementation by expression of the corresponding genes from separate plasmids fully restored conjugation proficiencies (**Fig 2A and 2B**). However, swaps of IMC genes between the pED208 and F systems failed to support DNA transfer at detectable levels (**Fig 2A and 2B**).

Previous studies have shown that TraD T4CPs are not required for elaboration of F or ED208 pili [46,47]. In contrast, the IMC deletion mutations abolished F and ED208 piliation, as evidenced by resistance of cells carrying the mutant plasmids to infection by male-specific bacteriophages (**Figs 2A, 2B and S3B**). The IMC deletion mutants were resistant to single-stranded (ss) RNA phage MS2, which binds the sides of F pili but not ED208 pili, as well as to ssDNA phage M13, which binds the tips of both F and ED208 pili [47,48]. We assessed MS2 sensitivity with a standard plaque assay, and sensitivity of non-lytic phage M13 using M13KO7, which confers $Kan^r$ upon infection. Complementation of the deletion mutations with the cognate IMC genes restored sensitivity to both phages in all cases, but complementation with the swapped IMC genes failed to restore phage sensitivity (**Figs 2A, 2B and S3B**). In

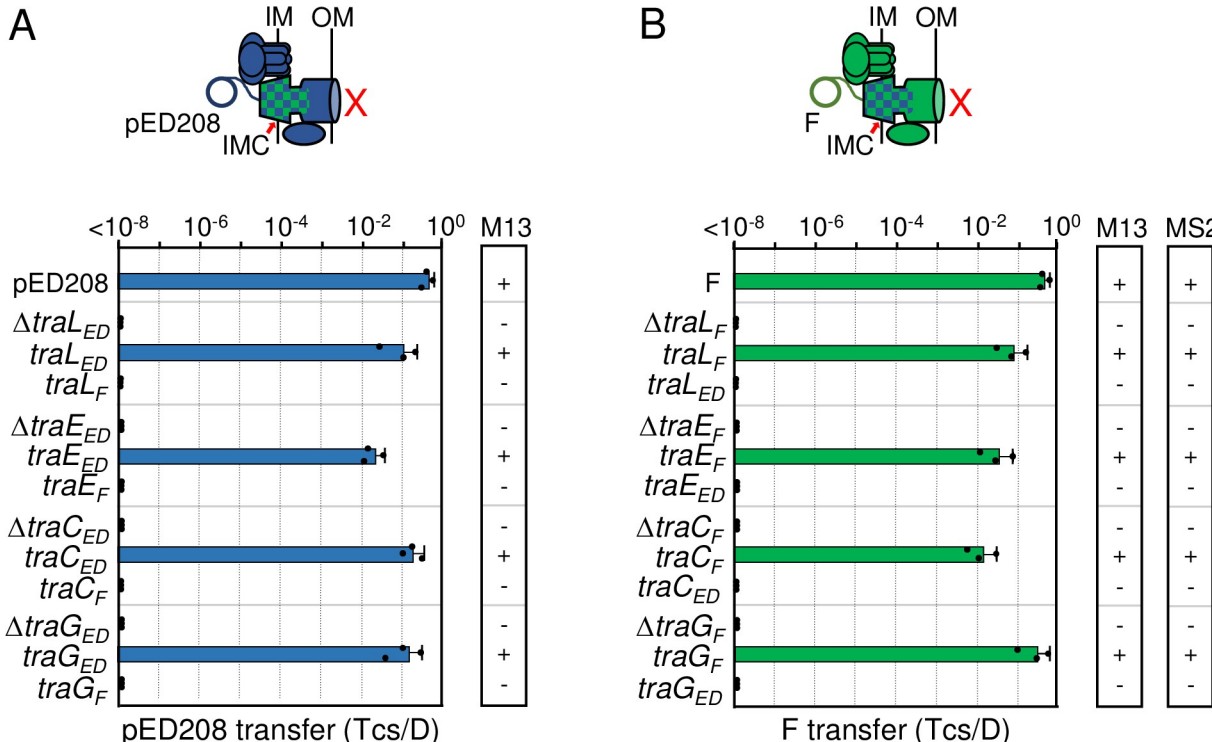

**Fig 2. F-like T4SSs with substituted IMC components are nonfunctional. A & B)** Upper: Schematics depicting chimeric T4SSs with substituted IMC subunits (blue/green checkered); these systems do not translocate substrates or elaborate pili (red X). Lower: pED208 or F transfer frequencies by donors bearing deletions of IMC genes or isogenic strains expressing the complementing cognate or noncognate genes. All matings were repeated at least three times in triplicate; a representative experiment is shown with replicate data points and the average transfer frequencies as horizontal bars along with standard deviations as error bars. M13 or MS2 columns: Susceptibility of donor cells to M13KO7 or MS2 phages: +, sensitive; -, resistant. M13KO7 quantitative data are presented in **S3 FigB** and source data in **S5 Table**.

both systems, therefore, the swapped IMC subunits either fail to integrate into the heterologous T4SSs or successfully integrate but render the chimeric T4SSs nonfunctional.

**Summary.** In the recently reported $T4SS_{R388}$ structure, components of the IMC were shown to form a highly-specific network of intersubunit interactions [5]. In a similar vein, the F-encoded IMCs might have evolved specific networks of intersubunit contacts, to the extent that subunit exchanges even of these structural homologs are not tolerated. In comparing the sequences of the Tra/Trb homologs between the pED208 and F systems, we noted that sequence divergence of several subunit pairs was confined to short stretches rather than distributed throughout the proteins. For the IMC homologs, this was particularly evident for the TraL subunits (**S1C Fig, red lines**). While such motifs might simply mark functionally-unimportant positions that accumulate mutations over time, they also may contribute to system-specific functions, for example, by mediating requisite contacts with cognate Tra subunits. Further mutation or swapping of such motifs between structural homologs should discriminate between these possibilities.

## F-encoded OMCCs tolerate subunit or domain deletions and swaps

No high-resolution structures exist for IMCs of F systems, but as mentioned earlier the $OMCC_{ED}$ structure was recently solved [14,15]. The maps reveal intriguing features, including the presence of two distinct substructures, the ORC and CC, which are only sparsely connected and have different rotational symmetries (**Fig 3A**). Remarkably, in testing the biological

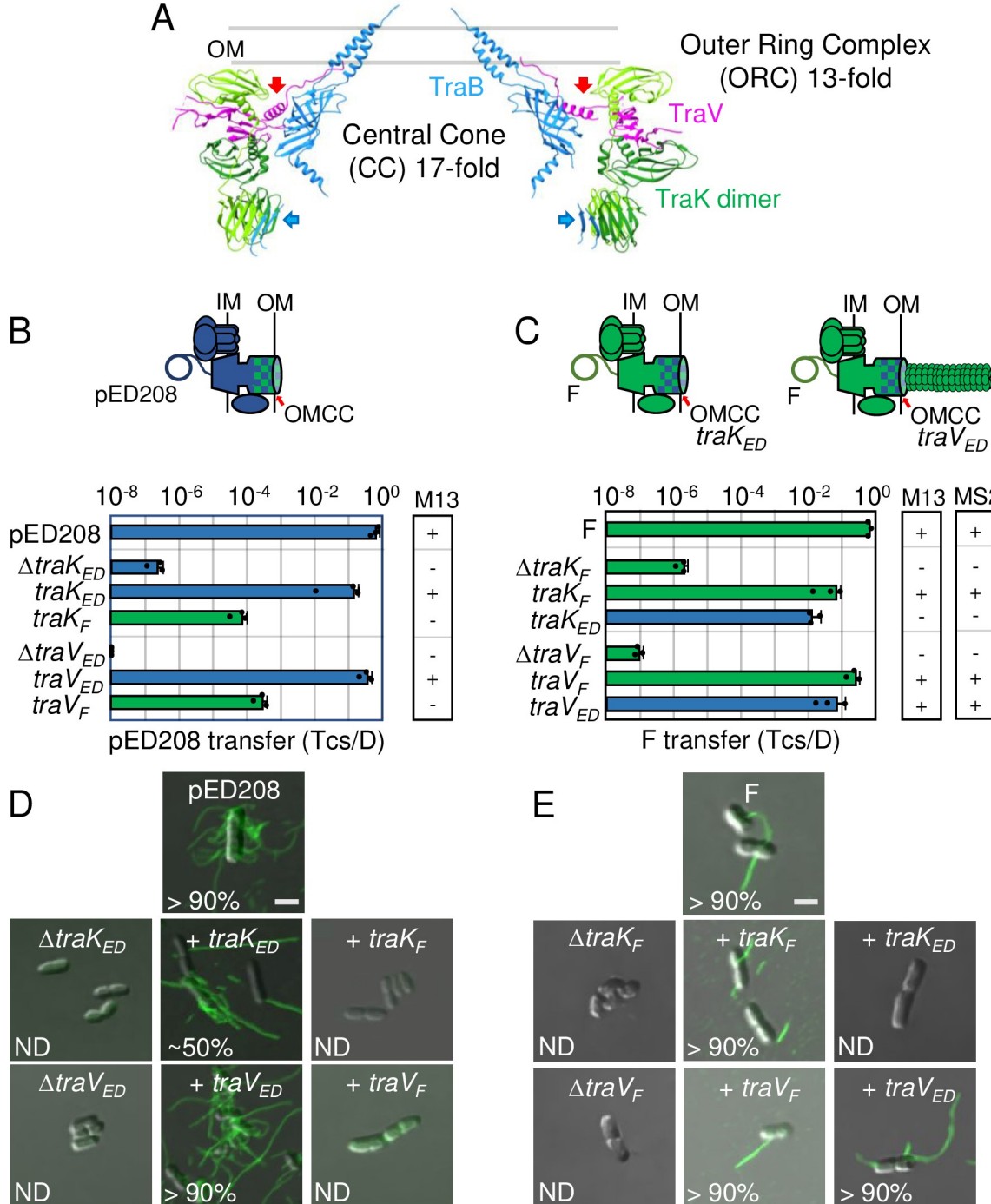

**Fig 3. Chimeric F-like T4SSs with substituted OMCC components can exhibit a Tra+, Pil- "uncoupling" phenotype. A)** A central slice of the atomic model of the pED208-encoded OMCC, generated using PDB files 7SPB and 7SPC and UCSF Chimera [49]. The central cone (CC) with 17-fold symmetry is composed mainly of the TraB β-barrel domains (blue); the TraB AP α-helical projections comprising the outer membrane (OM)-spanning channel are also shown. The outer ring complex (ORC) with 13-fold symmetry is composed of 13 TraK dimers (light and dark green shades) and 26 copies of TraV lipoprotein (pink). The CC and ORC are connected only by TraV (red arrows) and TraB (blue arrows) linker domains. **B & C)** Upper: Schematics depicting chimeric T4SSs with substituted OMCC subunits (blue/green checkered); these systems translocate substrates and may or may not elaborate pili. Lower: pED208 or F transfer frequencies by donors bearing deletions of OMCC genes or isogenic strains expressing the complementing cognate or noncognate genes. Plasmid transfer frequencies and phage susceptibilities are presented as described in Fig 2 legend. **D & E)** Visualization of Cys-derivatized ED208 pili by labeling with AF488-mal or F pili by labeling with MS2-GFP. Representative static images are shown for cells carrying pED208 or F, plasmid variants deleted of OMCC genes, or deletion plasmids along with plasmids

expressing the complementing cognate or noncognate genes shown. Numbers correspond to percentages of cells with detectable pili in the cell population; ND, none detected. Scale bars, 2 μm.

importance of various structural domains through mutational analyses, we discovered that the T4SS$_{ED}$ deleted of the major ORC component, TraK, retained the capacity to translocate pED208 but failed to elaborate the ED208 pilus [15].

**Deletions and swaps of ORC components confer "uncoupling" phenotypes.** To further interrogate the OMCC subunit and domain requirements for F-like systems, we first deleted and swapped ORC components (**Fig 3A**). Deletion of $traK_F$ phenocopied the $\Delta traK_{ED}$ mutation, rendering the T4SS$_F$ proficient for DNA transfer (~$10^{-6}$ Tcs/D) but unable to produce pili (**Fig 3B and 3C**). As our previous assays, e.g., phage infection, only indirectly tested for F pilus production, we additionally deployed a sensitive fluorescent labeling procedure that enabled direct visualization of ED208 pili [25]. We first engineered strains of interest to produce Cys-derivatized TraA$_{ED}$ pilins (TraA.C116) that support both pED208 transfer and ED208 pilus production at WT levels [25]. The Cys116 residues are surface-exposed on ED208 pili and accessible to labeling with fluorescent maleimide conjugates, e.g., FM488-mal. This method was recently shown also to label F pili [50], but here we deployed fluorescently-tagged ssRNA phage MS2, which binds the sides and readily labels these pili [25,47,51]. By fluorescent labeling of ED208 and F pili, we confirmed that cells carrying pED208$\Delta traK$ or F$\Delta traK$ indeed fail to elaborate pili despite their capacity to transfer plasmids (**Fig 3D and 3E**). The $\Delta traK$ mutant hosts also were resistant to infection by M13KO7, further showing that the $\Delta traK$ machines do not even produce 'vestigial' pili with only their tips exposed on the cell surface (**Figs 3D, 3E and S3C**). We term the $\Delta traK$ mutations as conferring a Tra+, Pil- "uncoupling" phenotype to reflect the fact they genetically uncouple the two major biogenesis pathways responsible for assembly of functional T4SS translocation channels vs conjugative pili.

A swap of TraK$_F$ for TraK$_{ED}$ elevated pED208 transfer by ~$10^{2.5}$-fold over that observed for the $\Delta traK_{ED}$ mutant machine, whereas the reciprocal swap elevated F transfer by ~$10^4$-fold, approaching frequencies observed for transfer through the native T4SS$_F$ (**Fig 3B and 3C**). Yet, neither chimeric machine elaborated detectable levels of pili (**Figs 3D, 3E and S3C**). These findings further support the notion that extracellular F pili are completely dispensable for efficient DNA transfer through F-like channels.

VirB7-like TraV subunits have N-terminal lipid modifications that serve OM as tethers, and an N-terminal α-helical domain (NTD) that binds laterally along the surface of the CC, forming contacts with three TraB β-barrel domains [5,15]. The C-terminal domain (CTD) assembles as two antiparallel β-strands connected by a loop. In the assembled OMCC$_{ED}$, the CTDs of two TraV monomers stack on top of each other, forming a 4-stranded β-sheet. These TraV CTDs connect laterally with adjacent TraV CTDs, so that collectively the 13 pairs of TraV CTDs build a belt that surrounds the 13 CTDs of TraK, presumably stabilizing the ORC substructure [15]. In the pED208- and F-encoded TraV subunits, the NTDs and CTDs are well-conserved, but the intervening linkers connecting the two domains are highly divergent (**S1D Fig**). This sequence divergence is in line with our previous finding that deletion of the TraV$_{ED}$ linker had no discernible effect on T4SS$_{ED}$ function [15]. Although we previously determined that the $\Delta traV_{ED}$ mutation blocked all T4SS$_{ED}$-associated activities [15], surprisingly the equivalent $\Delta traV_F$ mutation in the F system conferred the Tra$^+$, Pil$^-$ "uncoupling" phenotype (**Figs 3B–3E and S3C**).

A swap of TraV$_F$ for TraV$_{ED}$ in the pED208 system also conferred the "uncoupling" phenotype, as evidenced by restoration of pED208 transfer without comparable effects on ED208 pilus production. In striking contrast, the swap of TraV$_{ED}$ for TraV$_F$ fully restored both F transfer through the T4SS$_F$ and F pilus production (**Figs 3B–3E and S3C**).

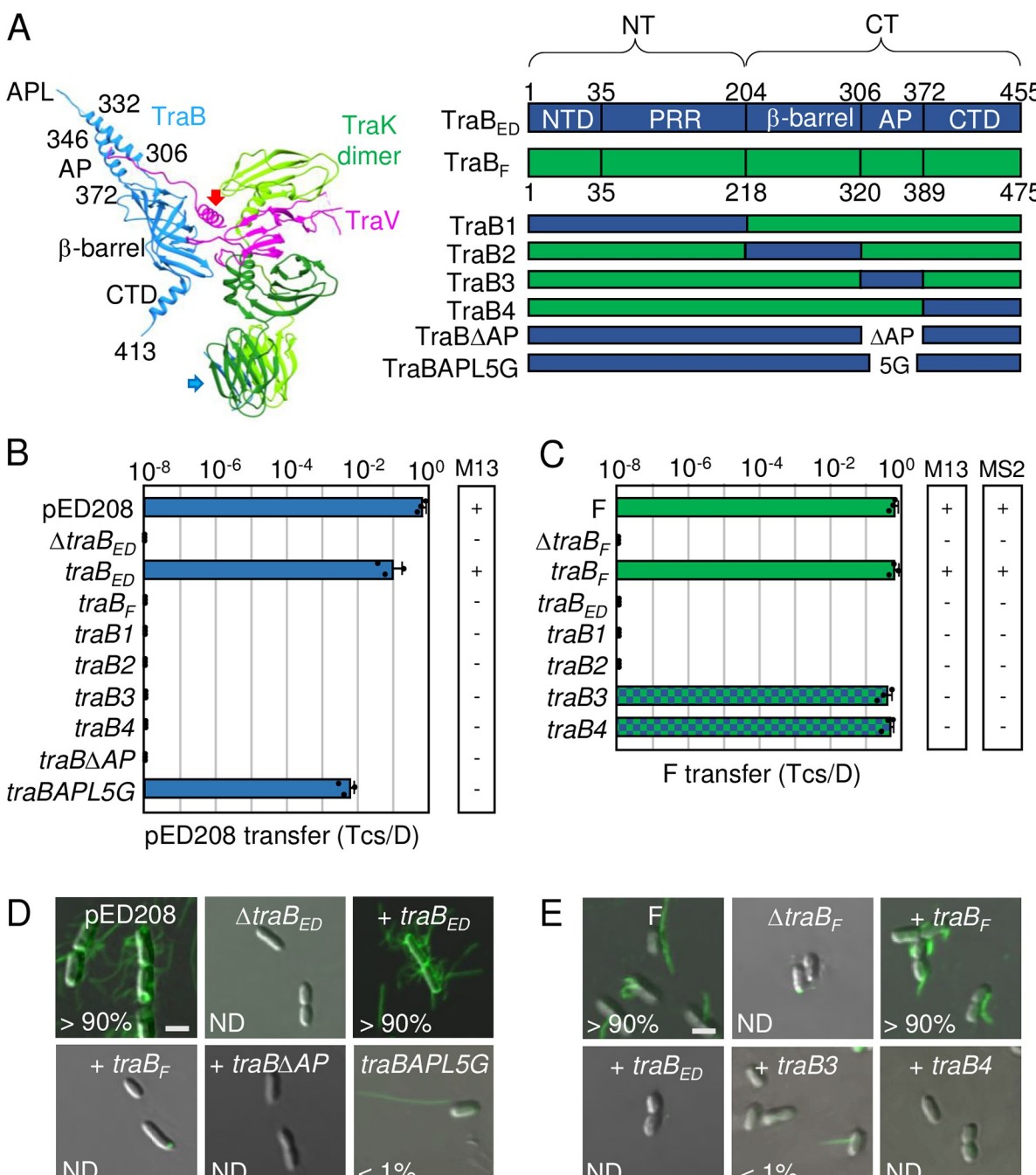

**Fig 4. F-like T4SSs with substituted TraB chimeras establish the functional importance of AP and C-terminal domains. A)** An asymmetric unit of the pED208-encoded OMCC generated as described in **Fig 3** legend. Thirteen TraK dimers (green shades) and C-terminal domains of TraV (pink) comprise the C13 ORC, and 17 TraB β-barrel (blue) and N-terminal domains of TraV (pink) assemble as the C17 CC. Flexible linkers of TraV and TraB bridge the ORC and CC substructures (red, blue arrows). Residues shown denote boundaries of the β-barrel, AP, and AP loop (APL) of TraB; the C-terminal domain (CTD) of TraB extending from residues 372–455 lines the surface of the β-barrel and extends into the lumen of the OMCC (unstructured). Right: Schematic of TraB homologs with domains shown as N-terminal transmembrane domain (NTD), proline-rich-region (PRR), β-barrel, antennae projection (AP), and C-terminal domain (CTD); numbers correspond to residues demarcating the domain boundaries. Schematics of TraB chimeras and mutants are depicted. **B & C)** pED208 or F transfer frequencies by donors bearing *traB* deletions or isogenic strains expressing the complementing genes shown. Plasmid transfer frequencies and phage susceptibilities are presented as described in **Fig 2** legend. **D & E)** Visualization of Cys-derivatized ED208 pili by labeling with AF488-mal or F pili by labeling with MS2-GFP. Representative static images are shown for cells carrying pED208 or F, or the corresponding Δ*traB* variants without or with the complementing genes shown. Numbers correspond to percentages of cells with detectable pili in the cell population; ND, none detected. Scale bars, 2 μm.

**Mutations and swaps of the AP and C-terminal domains of the CC component, TraB, confer "uncoupling" phenotypes.** Next, we tested whether the CC substructures tolerated swaps or mutations. The VirB10-like TraB subunits are multi-domain proteins composed of NTDs that span the inner membrane (IM), proline-rich regions (PRRs) that extend across the periplasm, and C-terminal β-barrel domains comprising the central cone or ring structures of OMCCs (**Fig 4A**) [2,52]. The $TraB_{ED}$ and $TraB_F$ homologs are only weakly related throughout their NTDs and PRRs, but their β-barrel domains are well conserved and have similar predicted structures (**S1D Fig**). A motif termed the antennae projection (AP; residues 306–372, $TraB_{ED}$ numbering) associates with the distal region of the TraB β-barrel (**Fig 4A**). This motif consists of two α-helices and an intervening loop (APL; residues 332–346). The 17 APs associated with the 17 β-barrels comprising the CC assemble as an OM-spanning channel, resulting in surface exposure of the APLs. The C-terminal ~80 residues (designated CTDs) of the 17 TraB subunits extend from the APs along the perimeter of the β-barrels and then project into the central chamber of the OMCC (**Fig 4A**) [5,15].

Swaps of the TraB homologs did not restore functionality of either T4SS (**Fig 4B–4E**), which was not surprising given that members of the VirB10 family span the entire cell envelope and their NTDs form various contacts with T4CPs and other IMC components [53–55]. To decipher contributions of specific TraB domains to T4SS functions, we constructed $TraB_F$ chimeras bearing the following $TraB_{ED}$ domains: NTD and PRR (designated TraB1), β-barrel (TraB2), AP (TraB3), or CTD (TraB4) (**Fig 4A**). All chimeras were stably produced (**S4B Fig**). The TraB1 and TraB2 chimeras failed to support plasmid transfer or pilus production in either the pED208 or F systems, showing that the N-terminal regions and β-barrel domains cannot be swapped (**Fig 4B and 4C**). The TraB3 (AP swap) and TraB4 (CTD swap) chimeras also failed to support pED208 transfer or ED208 pilus production (**Fig 4B and 4D**), but we attribute this to the nonfunctionality of $TraB_F$'s N-terminal region in the pED208 system. This is because, in the F system, the TraB3 and TraB4 chimeras supported F transfer through the $T4SS_F$ at WT levels, which establishes that the APs and CTDs indeed are swappable (**Fig 4C**). Remarkably, we detected F pili on <1% of cells carrying FΔ*traB* and producing the TraB3 chimera and no F pili on isogenic host cells producing TraB4 (**Fig 4E**). Consistently, both hosts also were resistant to M13KO7 and MS2 infection (**Fig 4C**). The $T4SS_F TraB3$ and $T4SS_F TraB4$ chimeric machines thus confer the Tra+,Pil- "uncoupling" phenotype.

We further interrogated the contribution of TraB's AP domain to T4SS function. To this end, we deleted the AP of $TraB_{ED}$ ($TraBΔAP_{ED}$) or substituted its APL with 5 Gly residues ($TraBAPL5G_{ED}$). Both variants were stably produced (**S4B Fig**). $TraBΔAP_{ED}$ failed to support ED208 transfer or pilus production (**Fig 4B and 4D**), establishing the essentiality of the OM channel for all $T4SS_{ED}$ functions. By contrast, $TraBAPL5G_{ED}$ supported pED208 transfer at high levels ($10^{-2}$ Tcs/D), but abrogated ED208 pilus production, as evidenced by M13KO7 resistance and detection of ED208 pili on <1% of cells examined (**Fig 4B and 4D**). The 5G motif evidently suffices for proper insertion of the AP α-helices into the OM as a prerequisite for substrate transfer; however, specific residues in the APL apparently are critical for pilus production.

In the above experiments, matings assessing the effects of OMCC mutations or subunit swaps were carried out under solid-surface conditions in which cells are densely packed. Remarkably, we also determined that nearly all strains bearing the Tra+,Pil- "uncoupling" mutations described above also were proficient as plasmid donors even in lower cell density aqueous conditions (**S4A Fig**). In general, these mutant donors transferred their plasmid cargoes in liquid matings at frequencies within 1–2 logs of levels reported for solid-surface matings. Although extracellular F-like pili clearly can enhance transfer of F-like plasmids, elaboration of these organelles is not required for functionality of F-encoded T4SSs and F plasmid dissemination even in dilute cell growth environments.

**Summary.**    Our findings underscore the remarkable plasticity of the F-encoded OMCCs in accommodating major structural perturbations without loss of DNA transfer functions. These perturbations include deletions of the TraK and TraV components of the ORC, swaps of both subunits, swaps of the AP and CT domains of TraB, and substitution of the APL with 5 Gly residues. Intriguingly, with the exception of the $TraV_{ED}$ swap for $TraV_F$ in the F system, all of these mutations conferred the Tra⁺, Pil⁻ "uncoupling" phenotype, as evidenced by resistance to male-specific phages and production of very few or no pili. Thus, with respect to pilus biogenesis, the F-encoded OMCC is considerably less flexible in accommodating the perturbations imposed here. To account for these findings at a mechanistic level, we propose that the subunit or domain swaps or mutations conferring the "uncoupling" phenotype disrupt system-specific contacts with other T4SS subunits that are selectively required for pilus production but not for substrate transfer. The peripheral ORC, as well as TraB's AP domain, are well-situated to form contacts with one or more F-specific proteins described below that might selectively regulate pilus production. On the other hand, both the surface-exposed APL and TraB's CTD positioned within the central chamber of the OMCC central chamber might contribute to piliation through direct interactions with TraA pilins or the growing pilus. Interestingly, the N- and C-terminal domains of the two TraV homologs, as well as the APL and CTD motifs of the TraB homologs, possess only a limited number of divergent sequences that might be responsible for the "uncoupling" phenotypes shown to accompany subunit swapping or mutation (**S1D Fig**).

We note that, while strains with the "uncoupling" phenotypes are devoid of extracellular pili, it remains likely that the TraA pilin is still needed as a component of the transfer channel. This proposal is supported by prior findings in the *Agrobacterium tumefaciens* VirB/VirD4 system, in which mutant strains exhibiting the Tra+,Pil- "uncoupling" phenotype still required production of the VirB2 pilin for detectable T-DNA transfer [56]. Conceivably, the IM-integrated forms of the VirB2/TraA-like pilins might comprise components of the IMCs required for substrate transfer across the IM. Alternatively, the pilins might be conveyed to the distal part of the T4SS to regulate an OM gate or assemble as a conduit for passage of the DNA substrate across the OM. With any of these scenarios, the Tra+, Pil- mutations would not impede functional integration of the pilins into the translocation channel but block steps in the pilus biogenesis pathway associated with pilus extension.

Perhaps most importantly, our findings strongly indicate that the F-like OMCCs are not simply passive players in F pilus biogenesis pathways, but rather actively participate in pilus assembly. This view is also supported by our recent *in situ* studies of the $T4SS_{ED}$, which supplied structural evidence that the F pilus nucleates assembly on an OM as opposed to an IM platform [46]. How the OMCC orchestrates nucleation of F pili and rounds of pilus extension and retraction remain intriguing questions for further study.

## Effects of F-specific subunit swaps

F-like plasmids encode at least eight F-specific subunits that enhance or are required for DNA transfer and pilus assembly. Recently, we characterized effects of F-specific gene deletions from pED208 and F, which resulted in assignments of the F-specific subunits into one of three classes: i) Class I factors (TraF, TraH, TraW) are required for DNA transfer and piliation, ii) Class II factors (TrbC, TraU, TraN) are not required but contribute significantly to both $T4SS_F$ functions, and iii) one Class III factor (TrbB) is essential for F pilus production but not for plasmid transfer [25]. One F-specific factor, TrbI, could not be classified because the Δ*trbI* mutations exerted polar effects on expression of other *tra/trb* genes. As shown for other Tra subunits, the F-specific homologs encoded by pED208 and F adopt similar predicted structural

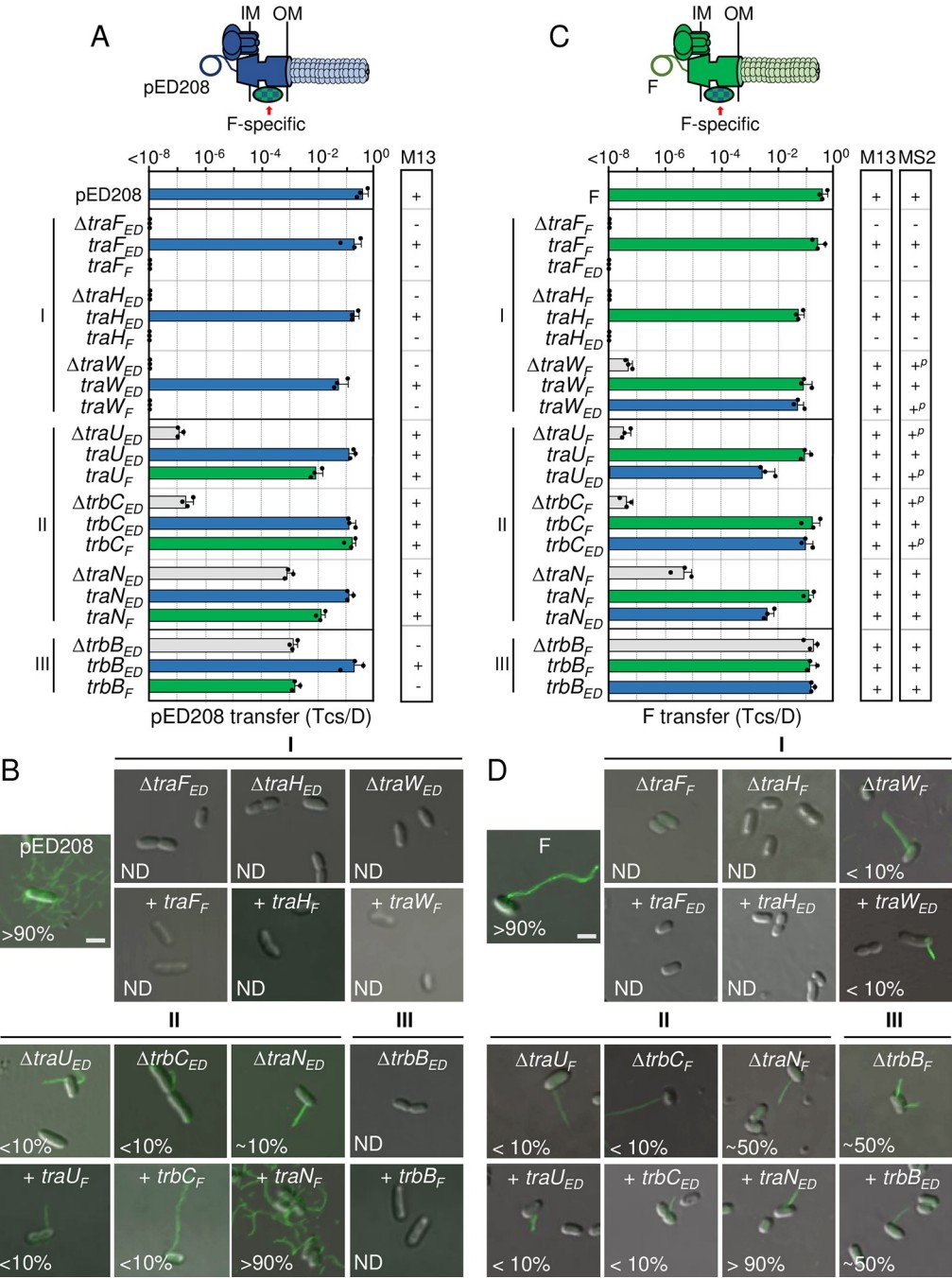

**Fig 5. Chimeric F-like T4SSs with substituted F-specific subunits confer distinct phenotypes. A & C)** Upper: Schematics depicting chimeric T4SSs with substituted F-specific subunits (blue/green checkered); these systems may or may not translocate substrates or elaborate pili (denoted as lighter coloration). Lower: pED208 or F transfer frequencies by donors bearing deletions of F-specific genes or isogenic strains expressing the complementing cognate or noncognate genes. F-specific subunits are grouped according to phenotypes of corresponding deletion mutations (Classes I, II, III, see [25]). Plasmid transfer frequencies are presented as described in Fig 2 legend. M13 or MS2 columns: Susceptibility of donor cells to M13 (M13KO7) or MS2 phages: +, sensitive; $+^p$, partially sensitive; -, resistant. M13KO7 quantitative data are presented in **S3D Fig** and source data in **S5 Table**. **B & D)** Visualization of Cys-derivatized ED208 pili by labeling with AF488-mal or F pili by labeling with MS2-GFP. Representative static images are shown for cells carrying pED208 or F, variants deleted of F-specific genes, or isogenic strains expressing the complementing cognate or noncognate genes. Numbers correspond to percentages of cells with detectable pili in the cell population; ND, none detected. Scale bars, 2 μm.

folds (**S1E Fig**). Sequence identities range from 33–68%, but the alignments show that a few F-specific homologs diverge in only a few short regions (**S1E Fig**) that might account for the observed phenotypes accompanying subunit swapping.

**Class I subunit swaps.** Deletions of TraF, TraH, or TraW rendered the pED208 system completely nonfunctional [25]. Here, we determined that swaps of F-encoded TraF, TraH, and TraW for their pED208 counterparts failed to restore functionality of the pED208 system, as shown by lack of DNA transfer, resistance to M13KO7 infection, and absence of ED208 pili (**Figs 5A, 5B** and **S3D**). The corresponding exchanges of $TraF_{ED}$ and $TraH_{ED}$ for their counterparts in the F system yielded similar outcomes (**Figs 5C, 5D** and **S3D**). In the F system, the $\Delta traW_F$ mutant strain retained a low level of DNA transfer, was at least partially sensitive to M13KO7 and MS2, and a few cells in the population (~5%) elaborated detectable pili (**Figs 5C, 5D** and **S3D**) [25]. Interestingly, the $TraW_{ED}$ swap for $TraW_F$ restored F transfer through the $T4SS_F$ nearly to WT levels, but did not enhance F pilus production or confer sensitivity to phages beyond levels observed with the $\Delta traW$ mutant (**Fig 5C and 5D**).

**Class II subunit swaps.** Deletions of the Class II factors (TraU, TrbC, TraN) strongly attenuated plasmid transfer and production of ED208 and F pili (**Fig 5A–5D**) [25]. Intriguingly, reciprocal swaps of TraU and TrbC in both systems restored transfer of pED208 and F nearly to WT levels without corresponding enhancement of pilus production). In contrast, both TraN swaps fully restored DNA transfer and pilus production in both systems to WT levels. These findings, and several other properties described below, distinguish TraN from the other Class II subunits and, indeed, all other Tra/Trb subunits.

**Class III subunit swaps.** TrbB, adopts a thioredoxin fold, possesses a catalytic CxxC motif, and functionally substitutes for the thioredoxin isomerase Dsb [57]. In the pED208 system, $TrbB_{ED}$ confers only a ~$10^2$-fold enhancement of DNA transfer, but is essential for pilus production (**Fig 5A and 5B**) [25]. Despite evidence that $TrbB_F$ can shuffle disulfide bonds even among substrates that are not involved in conjugation [57], the swap of $TrbB_F$ for $TrbB_{ED}$ failed to restore full function of the pED208 system (**Fig 5A and 5B**). Curiously, in the F system, $TrbB_F$ is completely dispensable for DNA transfer, but contributes quantitatively to F pilus production both in terms of the number of pili produced per cell and the number of cells in a population with pili [25]. The $TrbB_{ED}$ swap for $TrbB_F$ had no discernible effects on F plasmid transfer or the number of cells producing F pili relative to the $\Delta trbB_F$ mutant (**Fig 5C and 5D**).

**Summary.** The F-specific subunits have been proposed to assemble as one or F-specific complexes based on results of two-hybrid screens [44,58–60]. Notably, $TraF_F$ and $TraH_F$ interact with multiple F-specific partners as well as components of the OMCC, prompting models depicting these Class I factors as forming a central node that connects the F-specific complex (es) to the OMCC to regulate its activity [58–60]. Such a central role might explain why the TraF and TraH swaps failed to restore activity of the pED208 and F systems: the swapped subunits likely do not form the requisite contacts for functionality of the heterologous systems. TraW also was reported to interact with $TraF_F$ and $TraH_F$, as well as other F-specific subunits [58,60,61], which might account for its essentiality in the pED208 system. In the F system, however, $TraW_F$ appears to play a more peripheral role than $TraF_F$ or $TraH_F$, possibly because other interactions among the F-specific components compensate for its absence. Also of interest, $TraW_{ED}$ integrated functionally into the F system, restoring DNA transfer but not enhanced production of F pili. The central (residues ~70–100) and C-terminal (~140–218) regions of the TraW homologs (**S1E Fig**) diverge in their primary sequences; conceivably these regions carry motifs specifying binding partner interactions selectively required for pilus production.

The Class II factors, TraU and TrbC, also were shown to form contacts with other F-specific proteins [58,60–62]. Results of our swapping studies suggest that the pED208- and F-encoded

TrbC and TraU homologs share certain structural motifs or binding partner interactions, which enabled integration into the heterologous systems and restoration of DNA transfer. However, TrbC or TraU swapping did not restore pilus production, further implying that other motifs possibly within regions of sequence divergence orchestrate pilus production only in the context of the cognate T4SS. The TrbC homologs diverge in their sequences throughout their lengths, but the TraU homologs possess only a few small clusters of divergence (**S1E Fig**) that might mediate such system-specific contacts. Although phenotypes of the Δ*traN* mutations supported assignment of TraN as a Class II factor [25], TraN has a number of features that are unique among components of conjugation machines, including its integral OM topology [63,64]. TraN also has a large extracellular domain (ED) that promotes formation of mating pairs through binding of outer membrane proteins (OMPs) displayed by recipient cells [63,65–67]. Different TraN subunits elaborate structurally distinct ED's that bind different OMPs, which results in selective transfer of the TraN-encoding F plasmids to specific enterobacterial species. The ED of TraN$_F$, for example, mediates binding to *E. coli* OmpA, while the ED of TraN$_{ED}$ adopts a distinct structural fold consistent with binding of OmpW [24,65]. Thus, while TraN$_{ED}$ and TraN$_F$ are swappable between the pED208 and F systems, in fact, they mediate formation of stable mating pairs through distinct ligand—receptor interactions. TraN subunits also possess large, Cys-rich periplasmic regions that contribute in unspecified ways to DNA transfer and pilus production [25,68]. Intriguingly, no TraN interactions with other T4SS components have yet been identified [58,60], leaving open the question of how TraN$_F$ and TraN$_{ED}$ can function interchangeably in the pED208 and F systems.

The Class III factor, TrbB, is an enigma. Among the Tra/Trb proteins, TrbB subunits are the only catalytically-active thioredoxins, although both TraF and TrbC also adopt thioredoxin folds but lack the CxxC motifs required for activity [25,57,69]. TrbB exhibits disulfide isomerase activity [57], and thus has been proposed to shuffle disulfide bonds to promote correct folding of the many Tra/Trb subunits shown to possess multiple Cys residues (TraB-2, TraF-2, TraH-7, TraN-22, TraU-10, TrbC-2) [69]. In this context, it is remarkable that Δ*trbB* mutations selectively disrupt production of ED208 and F pili without commensurate effects on substrate transfer [25]. Moreover, TrbB subunit swapping failed to restore activities of the heterologous systems, despite the likelihood that both proteins possess similar catalytic functions. The data thus favor the notion that TrbB subunits establish system-specific partner interactions of critical importance for piliation but not translocation, independently of their disulfide isomerase activities. Such contacts might be mediated by sequence-divergent terminal regions of the TrbB homologs located outside of the thioredoxin folds (S1E Fig).

## TraA$_F$ pilins functionally integrate into the pED208 T4SS

The last Tra subunit required for assembly of F-encoded T4SSs is the TraA pilin. TraA is composed of four domains, with domains II and IV consisting of hydrophobic α-helices and domain III of a short stretch of basic residues (**S1F Fig**) [70,71]. Newly synthesized TraA propilins insert into the IM via domains II and IV, the long leader peptide is cleaved by leader peptidase I (LepB), and the resulting pool of IM-integrated pilins is used for reiterative cycles of F pilus extension and retraction [70]. In the assembled pilus, domains II and IV form the stacking interfaces of adjacent pilins, hydrophilic domain I and the extreme C terminus are surface-exposed, and domain III is located in the pilus lumen in a 1:1 stoichiometric association with IM-derived phospholipids [16].

The TraA$_{ED}$ and Tra$_F$ pilins are 44% identical, possess the same domain architecture, and adopt similar predicted tertiary structures (**S1F Fig**) as well as quaternary structures in the assembled ED208 and F pili [16]. Despite these similarities, reciprocal swaps of the TraA pilins

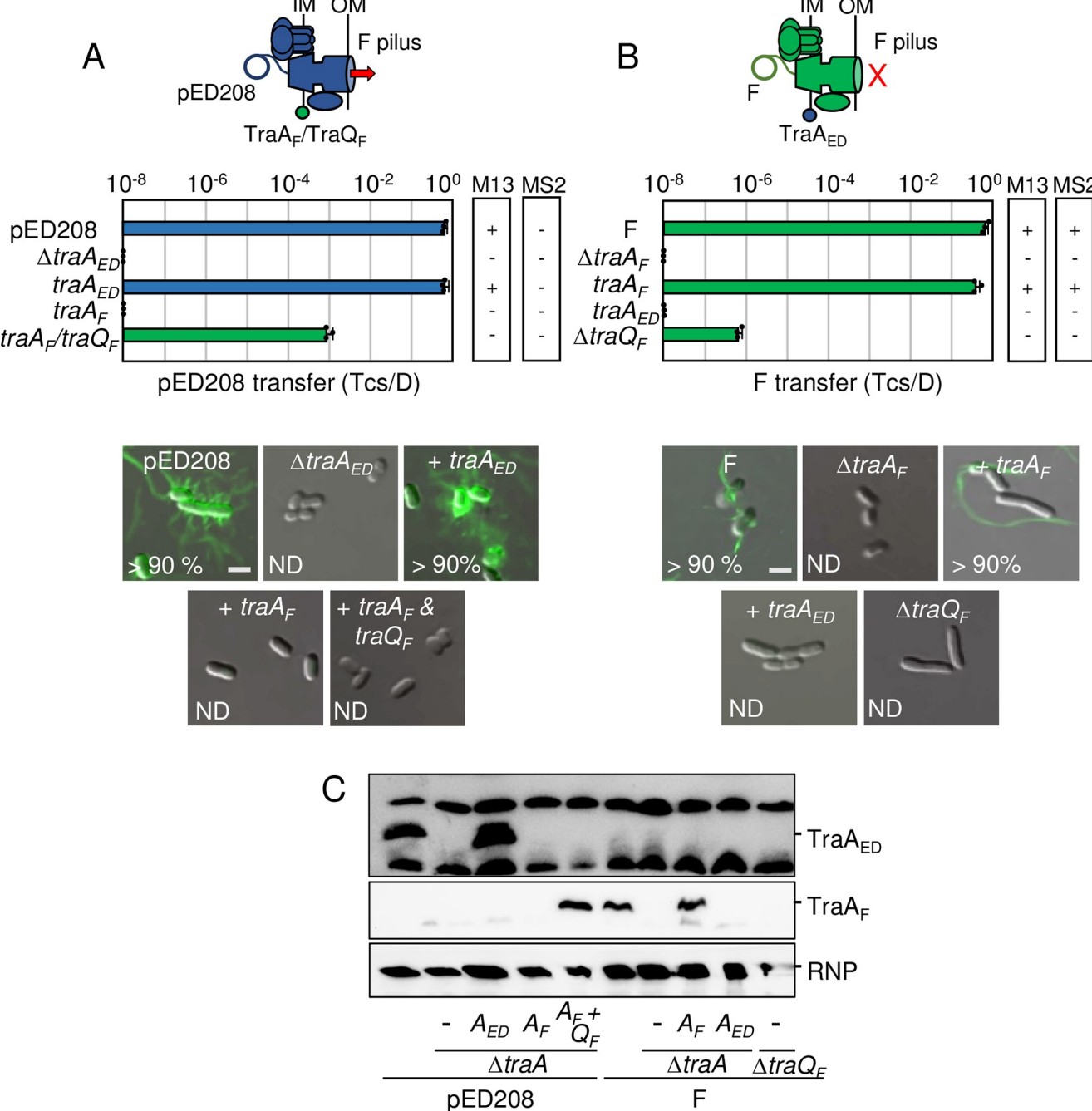

**Fig 6. A T4SS$_{ED}$TraA$_F$ chimeric machine translocates pED208 in the absence of F pilus production. A & B) Upper:** Schematics depicting T4SSs with substituted TraA pilin subunits (blue or green); the TraQ chaperone promotes integration of TraA$_F$ into the T4SS$_{ED}$ machine to support DNA transfer (red arrow) but not F pilus production. Middle: pED208 or F transfer frequencies by donors bearing Δ*traA* or *traQ$_F$* mutations, or isogenic strains expressing the complementing cognate or noncognate genes, without or with expression of the *traQ$_F$* chaperone. Plasmid transfer frequencies and phage susceptibilities are presented as described in Fig 2 legend. Lower: Visualization of Cys-derivatized ED208 pili by labeling with AF488-mal or F pili by labeling with MS2-GFP. Representative static images of cells carrying pED208 or F, Δ*traA* variants, or isogenic strains expressing the gene shown. Numbers correspond to percentages of cells with detectable pili in the cell population; ND, none detected. Scale bars, 2 μm. **C)** Immunoblotting of cellular fractions for detection of TraA$_{ED}$ or Tra$_F$ pilin subunits. Blots were developed with α-TraA antibodies specific for the pED208-encoded or F-encoded TraA pilins, and with antibodies against *E. coli* RNA polymerase β-subunit (RNP) as a loading control. The upper and lower bands in the TraA$_{ED}$ blot are nonspecific proteins reactive to the polyclonal α-TraA$_{ED}$ antibodies. Strains carried pED208 or F, or the corresponding Δ*traA* variants without or with plasmids expressing the *traA* pilin or *traQ* chaperone genes shown.

failed to restore T4SS$_{ED}$ or T4SS$_F$ functions (**Fig 6A and 6B**). We discovered one reason for the lack of cross-functionality, namely, host strains carrying pED208Δ*traA* and expressing *traA$_F$* or FΔ*traA* and expressing *traA$_{ED}$* failed to accumulate either of the TraA pilins at detectable levels (**Fig 6C**). These findings suggested that the pED208 and F systems lack factors respectively required for stabilization of TraA$_F$ and TraA$_{ED}$. In the F system, the membrane-bound chaperone TraQ was shown to direct insertion and stabilization of TraA$_F$ in the IM [72, 73]. As pED208 lacks a discernible *traQ* gene [24], we coexpressed *traA$_F$* and *traQ$_F$* in a strain harboring pED208Δ*traA*. Indeed, this strain accumulated abundant levels of TraA$_F$ (**Fig 6C**), and efficiently transferred the pED208 plasmid to recipients (**Fig 6A**). This strain also delivered its plasmid cargo to recipients in liquid matings, albeit at a frequency of $\sim 10^{-3}$-fold less than achieved with solid-surface matings (**S4A Fig**). Remarkably, this strain failed to elaborate F pili as shown by resistance to both M13KO7 and MS2 phages and a lack of detectable pilus labeling (**Figs 6 and S3E**). The T4SS$_{ED}$TraA$_F$/TraQ$_F$ chimeric machine thus confers the Tra+, Pil- "uncoupling" phenotype.

We envisioned that another pED208-encoded factor might supply the TraQ-like chaperone function necessary for stabilization of TraA$_{ED}$ in the IM. We analyzed our collection of pED208 *tra/trb* mutations for effects on TraA$_{ED}$ production, but all mutant strains accumulated the pilin at detectable levels (**S4C Fig**). pED208 carries several uncharacterized *orfs* in the *tra* region that might encode a TraQ-like function. Three (*orfX1-3*) are embedded in the OMCC gene cluster and one (*orfX4*) is in the F-specific gene cluster, which is where *traQ* resides in F (**S4D Fig**) [24]. We deleted the OMCC and F-specific gene clusters from pED208, but strains carrying the resulting plasmids also accumulated TraA$_{ED}$ pilin at abundant levels, arguing against contributions of the uncharacterized *orfs*, the entire OMCC, or F-specific complexes to TraA$_{ED}$ stabilization (**S4D Fig**). It remains possible that a factor encoded elsewhere on pED208 is required for stabilization of TraA$_{ED}$.

**Summary.**   Presently, it is not known how pilin subunits of T4SSs contribute to channel assembly or activity, or where they nucleate to build the extracellular filament. We recently solved *in situ* structures of the pED208-encoded channel without and with the associated ED208 pilus in the native environment of the bacterial cell envelope [46]. Several features of these structures support a model depicting the assembly of F-like pili on a platform near or at the OM. In the solved structure of the T4SS$_{ED}$ channel, there is a clearly defined cylindrical tube that extends from the IM and projects up through the OMCC, but does not cross the OM. Intriguingly, this central tube is present even in the solved structure of a Δ*traA$_{ED}$* mutant machine, establishing that it is not built from TraA$_{ED}$ pilins [46]. Moreover, in the structure of the T4SS$_{ED}$ channel associated with the ED208 pilus, the pilus attaches to the OMCC at the OM junction, but several densities of unknown composition block the pilus lumen from joining with the cylindrical tube in the periplasm [46]. Here, our finding that TraA$_F$ integrates into the pED208 system to restore DNA transfer without elaborating F pili, strongly indicates that TraA$_F$ pilins engage productively with the T4SS$_{ED}$ channel, but fail to form interactions with other T4SS subunits necessary for nucleation of the pilus. As mentioned earlier, the IM-integrated pool of TraA$_F$ pilins might assemble as an essential part of the IMC or IM-extracted pilins might be conveyed to the distal region of the channel to regulate an OM gate or assemble as a conduit across the OM. Favoring this latter possibility, studies have now firmly established that F-carrying donors are capable of delivering plasmid substrates to distant recipients [74] through the F pilus [50,75]. These are rare events [50] and likely occur only in certain circumstances, for example, when donor and recipient cells are physically constrained from being drawn together through F pilus retraction. Through sensing of an obstruction in pilus retraction, the T4SS$_F$ might default to an alternative pathway by translocating the DNA transfer intermediate through the extended pilus. Structurally, this could imply that F pili correspond

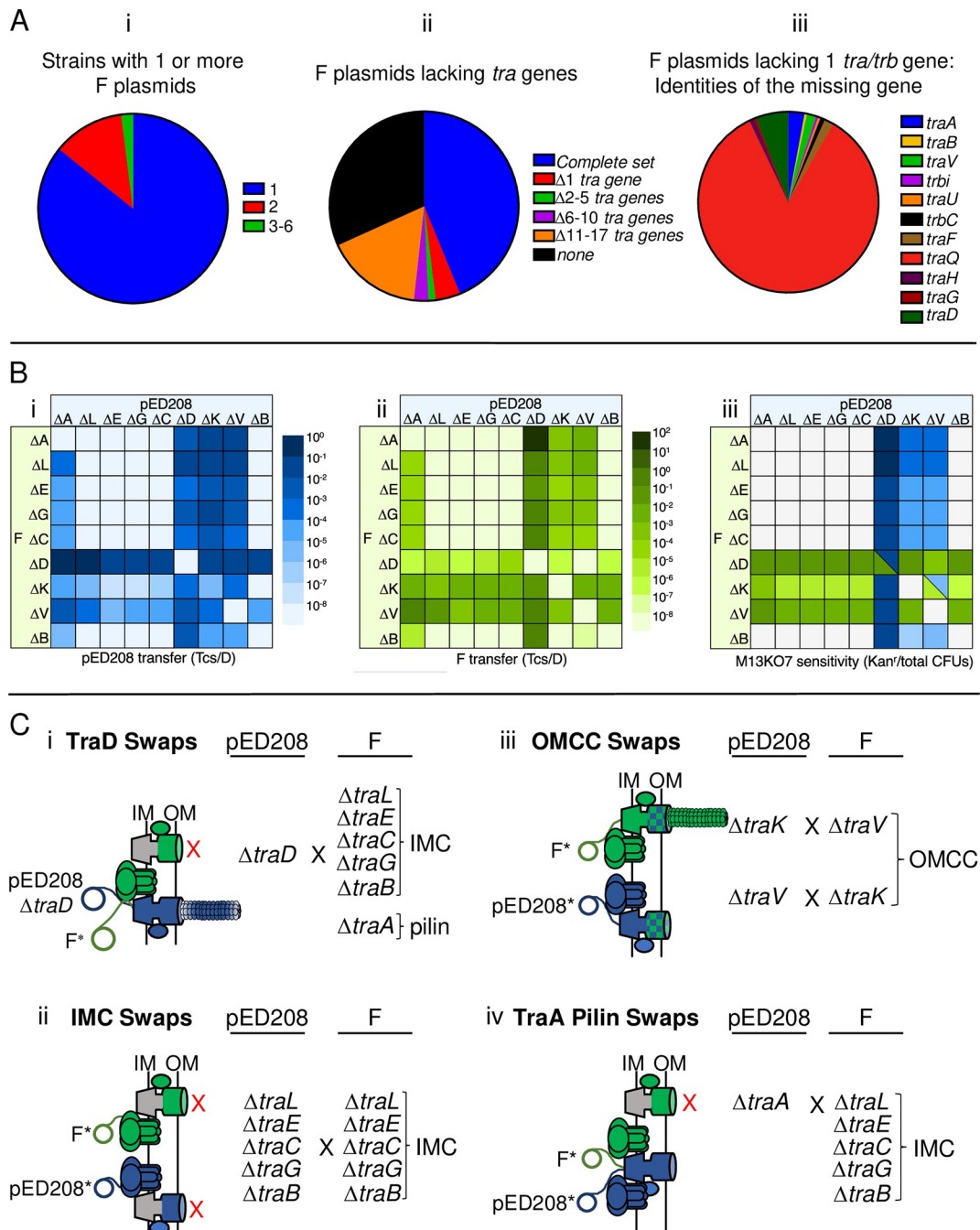

**Fig 7. Strains with coresident, transfer-defective pED208 and F plasmids assemble functional T4SS chimeric machines.**
**A)** Pie charts showing prevalence of F plasmids and of *tra/trb* genes that are missing among F plasmids sourced from the
COMPASS database [76] and PLSDB (v. 2021_06_23_v2) [79]. **i)** The number of F plasmids per strain among all strains
shown to carry at least one F plasmid. Colors represent the number of F plasmids per strain. A total of 908 strains carrying at
least one F plasmid were identified and analyzed. **ii)** The prevalence of F plasmids with completely intact or missing *tra/trb*
genes. Colors represent the fraction of F plasmids with the depicted number of missing *tra/trb* genes. A total of 5,664 F
plasmid sequences were analyzed from PLSDB (v. 2021_06_23_v2). The tblastn searches were performed against these F
plasmid sequences using Tra and Trb proteins encoded by the following F plasmids as queries: pOX38 (MF370216.1),
pED208 (AF411480.1), pKpQIL-UK (KY798507.1), pOZ172 (CP016763.1) (see Materials and methods). **iii)** The fractions of
specific *tra/trb* genes that were missing among F plasmids shown to lack one *tra/trb* gene. Source data for panels Ai, Aii, and
Aiii appear in S3 Table. **B)** Heat maps showing transfer frequencies of **Bi)** pED208 (blue) and **Bii)** F (green) variants deleted
of the *tra* genes shown by donor strains carrying the pairwise combinations of mutant plasmids depicted. Transfer
frequencies (Tcs/D) are color-graded as depicted. **Biii)** M13KO7 susceptibility of host strains carrying the mutant plasmids

depicted. M13KO7 susceptibilities (Kan$^r$ colonies/total CFUs) are color-graded as depicted. Quantitative data for panels Bi, Bii, and Biii are presented in **S4 Table** and source data in **S5 Table**. The color shades reflect our predictions of what pili are produced enabling M13 infection; blue, ED208 pili; green, F pili; blue/green triangles, both pili are produced. **C)** Schematics and summaries of transfer and piliation among donors co-harboring pED208 and F plasmids with the mutations indicated. Strains with co-resident mutant plasmids are grouped to highlight: **Ci)** exchangeability of TraD T4CPs, **Cii)** lack of functional exchanges of IMC subunits, **Ciii)** exchangeability of OMCC subunits TraV and TraK, **Civ)** functional incorporation of *traA$_F$* into the pED208 system, resulting in the Tra$^+$, Pil$^-$ "uncoupling" phenotype.

to extracellular extensions of the TraA OM conduit that normally functions to convey substrates across the OM once F pili have retracted, or when pilus extension is entirely blocked through mutation. Further mutational studies of the divergent sequences between the TraA$_{ED}$ and TraA$_F$ homologs (**S1F Fig**) should unveil domains or residues required for the distinct activities of these pilins in controlling channel function vs piliation.

## Strains harboring co-resident pED208 and F mutant plasmids naturally assemble functional chimeric machines

Our findings to this point established that swaps of several *tra/trb* genes yielded chimeric machines functional at least for DNA transfer. The potential for chimeric T4SSs to assemble in natural settings is medically important in view of recent evidence that enterobacterial clinical isolates can carry two or more F-like plasmids [76–78]. Here, we surveyed the COMPASS database [76] for strains carrying one or more F plasmids, and determined that indeed an appreciable fraction of sequenced strains (~15%) harbor two or more F plasmids (**Fig 7Ai**), Recently, it was also reported ~25% of 'classical' F plasmids (those designated as group A) lack one or more genes required for F transfer [26]. We extended this analysis to include 5,664 F-like plasmids in the COMPASS database [76], which surprisingly showed that many are missing one or more *tra/trb* genes (**Fig 7Aii**). Among the F-like plasmids lacking one *tra* gene, ~85% are missing *traQ* (**Fig 7Aiii**). Like pED208, these F-like plasmids might rely on another chaperone T4SSs for pilus biogenesis. However, among the remaining mutant F plasmids, many lacked genes encoding functions shown here to be complementable by subunit swapping between the distantly related F and pED208 systems (**Fig 7Aiii**).

In view of these findings, we hypothesized that strains with multiple F plasmids might disseminate them widely even if one or more are non-selftransmissible. To test this idea, we constructed *E. coli* donors with all combinations of pED208 and F variants deleted of genes for the core VirB- and VirD4-like functions. We then assayed for transfer of both plasmids and at least the production of vestigial pili as monitored by M13KO7 phage sensitivity. The DNA transfer and phage infection data are presented in **Fig 7B**, and a subset of the notable findings are summarized schematically in **Fig 7C**.

Overall, results were consistent with data presented for the *tra/trb* gene swaps. For strains harboring one Δ*traD* mutant plasmid, for example, both plasmids were conjugatively transferred even in cases when the second plasmid carried mutations, e.g., IMC deletions, that rendered the encoded T4SS nonfunctional (**Fig 7B and 7Ci**). Moreover, both plasmids were transferred at frequencies expected of the chimeras predicted from results of our TraD swapping studies. Strains capable of assembling the T4SS$_F$TraD$_{ED}$ chimera transferred pED208 variants at high frequencies ($10^{-1-2}$ Tcs/D), but F variants at low frequencies ($10^{-5-6}$ Tcs/D) because TraD$_{ED}$ functions poorly in recruitment of the F substrate (see **Fig 1**). Conversely, strains competent for assembly of T4SS$_{ED}$TraD$_F$ transferred both plasmids at high frequencies because TraD$_F$ efficiently recruits both F and pED208 substrates (see **Fig 1**). Because TraD does not contribute to production of F-like pili [47], strains harboring at least one Δ*traD* mutant plasmid elaborate pili and thus were sensitive to M13KO7 infection (**Fig 7Biii**). A

strain carrying pED208Δ*traD* and FΔ*traD* is transfer-defective but M13KO7 sensitive, the latter presumably due to production of both ED208 and F pili (**Fig 7Biii**).

Strains with combinations of pED208 and F plasmids deleted of IMC genes were invariably transfer-minus and resistant to M13KO7 infection (**Fig 7B and 7Cii**). These findings strengthen our proposal that the IMC homologs cannot be exchanged between these systems due to the evolution of system-specific interaction networks among IMC substructures.

With respect to the OMCC components, strains with co-resident plasmids bearing combinations of Δ*traK* and Δ*traV* mutations transferred both plasmids at moderately high frequencies of ~$10^{-4}$ Tcs/D (**Fig 7B and 7Ciii**). These frequencies were considerably higher than the transfer frequencies observed for T4SSs lacking TraV or TraK ($10^{-8-6}$ Tcs/D) (**Fig 3B and 3C**), confirming that TraV and TraK are freely exchanged even when the cognate machines are produced. Remarkably, nearly all strains carrying pED208Δ*traK* and an F plasmid capable of donating TraK to the T4SS$_{ED}$ were sensitive to M13KO7 infection. Similarly, strains carrying pED208Δ*traV*, FΔ*traK*, or FΔ*traV* along with another F or pED208 plasmid capable of donating the missing TraV or TraK subunits were M13KO7 sensitive (**Fig 7Biii**). These findings contrast with our earlier results showing that, among the combinations of OMCC swaps, only the exchange of TraV$_{ED}$ for TraV$_F$ in the F system supported production of F pili (**Fig 3E**). Insofar as TraV or TraK assemble as unusual dimers when incorporated as the structural building blocks of the ORC, one explanation for these findings is that these subunits are more readily exchanged when they are first able to fold and form stable dimers in the context of their cognate machines. Regardless of the underlying mechanism, the finding that strains with combinations of F-like variants harboring OMCC mutations are minimally capable of elaborating 'vestigial' pili is suggestive of a functional synergism between coresident mutant machines.

Finally, strains carrying pED208Δ*traA* and transfer-defective F variants, e.g., IMC deletion mutants, were proficient for transfer of both plasmids (**Fig 7B and 7Civ**). These results are consistent with our findings that the pED208 system is capable of recruiting TraA$_F$ pilins from a TraQ$_F$-stabilized IM pool (**Fig 6A**). Additionally, strains capable of assembling T4SS$_{ED}$TraA$_F$ chimeric machines were resistant to M13KO7 infection (**Fig 7Biii**), showing that these chimeras fail to elaborate F pili or display pilus tips at least in the native configuration on the cell surface. In contrast, strains carrying FΔ*traA* and transfer-defective pED208 variants failed to transfer either plasmid (**Fig 7B**), which also is consistent with our above findings that TraA$_{ED}$ pilins fail to integrate productively into the F-encoded T4SS (see **Fig 6B**).

**Overall conclusions: T4SS chimeras constitute a mechanism for functional diversification and MGE dissemination.** By systematically analyzing effects of subunit or domain exchanges between distantly-related F and pED208 systems, we identified regions of the encoded T4SSs that accommodate sequence or structural perturbations without loss of DNA transfer proficiency. Remarkably, many of the chimeric machines were deficient in pilus production, suggesting that the requirements for elaboration of dynamic F pili are considerably more constrained than for elaboration of functional translocation channels. Mechanistically, the ability to genetically "uncouple" pathways for elaboration of functional channels vs biogenesis of F pili sets the stage for further definition of structural features and Tra/Trb partner interactions required for one but not the second of these dynamic processes. On an evolutionary scale, it is reasonable to propose that intrinsic flexibility at the substrate/T4CP and T4CP/T4SS interfaces has contributed to diversification of T4SS substrate repertoires. Plasticity of the OMCC also can be predicted to account for the remarkable structural diversity recently shown to exist among OMCCs [2]. Although it is hypothesized that this structural diversity endows T4SSs with specialized functions, such structure—function relationships await definition. On a more immediate time scale, the capacity of co-resident MGEs to build functional

chimeric systems offers a general mechanism that can account for the persistence and spread of non-selftransmissible elements in natural settings. That many of the F-like chimeras characterized here were proficient in DNA transfer but not in pilus production suggests that T4SS chimerism has a further benefit for both the bacterial host and resident conjugative plasmids. While such strains are proficient for dissemination of MGEs and associated fitness traits, they also are resistant to host cell lysis and plasmid loss resulting from infection by male-specific phages.

## Materials and methods

### Strains and growth conditions

*E. coli* strains listed in **S1 Table** were grown in Lysogeny Broth (LB) at 30˚C and 42˚C for recombineering and 37˚C for other applications. Strains and plasmids were maintained with antibiotic selection as appropriate: spectinomycin (100 μg ml$^{-1}$), carbenicillin (100 μg ml$^{-1}$), kanamycin (100 μg ml$^{-1}$), tetracycline (20 μg ml$^{-1}$), chloramphenicol (20 μg ml$^{-1}$), and rifampicin (100 μg ml$^{-1}$).

### Plasmid constructions

**F and pED208 mutations.**   All plasmids and oligonucleotide primers used in these studies are listed in **S1 and S2 Tables**. *E. coli* strain HME45 were used to generate deletions of the *tra* genes from pED208 or pOX38 (the F plasmid) by recombineering as previously described [25,80]. A FRT-Kan$^r$-FRT cassette from pKD13 was amplified with primers listed in S**2** Table to carry homologous sequence to the upstream and downstream regions of the region targeted for deletion. The λ *red-gam* system in HME45 cells carrying pED208 or F was induced by growth at 42˚C, and the FRT-Kan$^r$-FRT amplicon of interest was introduced by electroporation, with Kan$^r$ selection for recombinants. Complementing plasmids were introduced into recombinant strains for conjugative transfer of the pED208 or F plasmids harboring FRT-Kan$^r$-FRT cassettes into MC4100 cells carrying pCP20, which expresses the Flp recombinase. Transconjugants were grown at 42˚C overnight to induce Flp recombinase expression for excision of the FRT-Kan$^r$-FRT cassette and to cure pCP20. *E. coli* strain MC4100 carrying pKD46 were used to generate deletions of the F-specific or OMCC gene clusters from pED208 or F. The λ *red-gam* system from pKD46 cells carrying pED208 or F was induced with 0.2% arabinose, and the FRT-Kan$^r$-FRT amplicon of interest was introduced by electroporation, with Kan$^r$ selection at 42˚C to generate recombinants and cure cells of pKD46. pCP20 were introduced into the recombinant strains for excision of the FRT-Kan$^r$-FRT cassette. Substitutions of *tra/trb* genes with the FRT scar were confirmed by sequencing across the recombination junctions.

**Cloning of *tra/trb* genes.**   The pED208 and F *tra/trb* genes were amplified by PCR using primers listed in **S2 Table**. Amplicons were digested with NheI and HindIII, and the resulting fragments were inserted into similarly digested pBAD24. Plasmid pKKF083 expressing *traA$_F$* from the *nahG* promoter was constructed by PCR amplification, digestion of the PCR product with NdeI and BamHI, and insertion of the resulting fragment into similarly digested pKG116. Plasmids pKN9—pKN12 expressing different *traB* chimeras were constructed by Gibson assembly (NEB). Partial *traB* DNA fragments were amplified by PCR using pED208 or F as a template and primers listed in **S2 Table**. The resulting fragments were inserted into pBAD24 digested with NheI and HindIII by Gibson assembly.

***traD*, *traJ*, and *traB* constructions.**   pYGL343 and pYGL351 expressing N-terminally Strep-tagged *traD$_{ED}$* and *traDΔC15$_{ED}$*, respectively, from the *nahG* promoter were constructed by PCR amplification using pED208 as the template, digestion of the PCR products with NdeI and BamHI, and insertion of the resulting fragments into similarly digested pKG116.

pYGL342 and pYGL348 expressing N-terminally Strep-tagged $traD_F$ and $traD\Delta C15_F$, respectively, from the *nahG* promoter were constructed by PCR amplification using pOX38 as the template, digestion of the PCR products with NdeI and BamHI, and insertion of the resulting fragments into similarly digested pKG116. pYGL491 expressing $traD\Delta C166_{ED}$ from the *nahG* promoter was constructed by deleting the C166 sequence from pYGL343 by inverse PCR. pYGL492 expressing and $traD\Delta C148_F$ from the *nahG* promoter was constructed by deleting the C148 sequence from pYGL342 by inverse PCR. pYGL353 expressing $traD_{ED}C15_F$ from the *nahG* promoter was constructed by changing the C15 sequence of pYGL343 by inverse PCR. pYGL352 expressing $traD_FC15_{ED}$ from the *nahG* promoter was constructed by changing the C15 sequence of pYGL342 by inverse PCR. pYGL511 expressing $traD_{ED}C148_F$ from the *nahG* promoter was constructed by PCR amplifications using pYGL343 and pOX38 as templates, digestion of the two PCR products with KpnI and HindIII, and ligation of the resulting products. pYGL510 expressing $traD_FC166_{ED}$ from the *nahG* promoter was constructed by PCR amplifications using pYGL342 and pED208 as templates, digestion of the two PCR products with KpnI and HindIII, and ligation of the resulting products. pYGL493 expressing Strep-tagged $traJ_{KM}$ from the *nahG* promoter was constructed by PCR amplification using pKM101 as the template, digestion of the PCR products with NdeI and BamHI, and insertion of the resulting fragments into similarly digested pKG116. pYGL494 expressing $traJ_{KM}C166_{ED}$ from the *nahG* promoter was constructed by PCR amplifications using pKM101 and pYGL343 as templates, digestion of the two PCR products with NdeI and KpnI, and ligation of the resulting products. pYGL528 expressing $N1\text{-}134_{ED}traJ_{KM}$ from the *nahG* promoter was constructed by PCR amplifications using pKM101 and pYGL491 as templates, digestion of the two PCR products with NsiI and SpeI, and ligation of the resulting products. pYGL529 expressing $N1\text{-}134_{ED}traJ_{KM}C166_{ED}$ from the *nahG* promoter was constructed by PCR amplifications using pKM101 and pYGL343 as templates, digestion of the two PCR products with NsiI and SpeI, and ligation of the resulting products.

pPK020 expressing $traB\Delta AP_{ED}$ from the $P_{BAD}$ promoter was constructed by inverse PCR amplification using pPK019 as the template and ligation of the resulting product. pPK021 expressing $traBAPL5G_{ED}$ from the $P_{BAD}$ promoter was constructed by inverse PCR amplification using pPK019 as the template and ligation of the resulting product. pYGL683 and pYGL684 expressing $traB\Delta Ap_{strep}$ and $traBAPL5G_{strep}$, respectively, from the $P_{BAD}$ promoter were constructed by PCR amplification using pPK020 and pPK021 as the templates respectively, digestion of the PCR products with NheI and HindIII, and insertion of the resulting fragments into similarly digested pBAD24.

**p*oriT* plasmid constructions.**   pCGR97 carrying the pKM101 *oriT* sequence and expressing $traK\text{-}traJ\text{-}traI_{KM}$ was constructed by PCR amplification using pKM101 as the template, digestion of the PCR products with NheI and HindIII, and insertion of the resulting fragments into similarly digested pBAD24. pYGL490 carrying the pKM101 *oriT* sequence and expressing $traK\text{-}traI_{KM}$ was constructed by deleting *traJ* from pCGR97 by inverse PCR. pYGL248 carrying the F *oriT* sequence was constructed by PCR amplification using pOX38 as the template, digestion of the PCR products with NotI and HindIII, and insertion of the resulting fragments into similarly digested pBAD101. pYGL249 carrying the pED208 *oriT* sequence was constructed by PCR amplification using pED208 as the template, digestion of the PCR products with NotI and HindIII, and insertion of the resulting fragments into similarly digested pBAD101.

## Conjugation assays

Donor and recipient cells were grown overnight at 37°C in presence of the appropriate antibiotics, diluted 1:50 in fresh antibiotic-free LB media, and incubated with shaking for 1.5 h. For

solid-surface matings, 10 μl of donor and recipient cell cultures were mixed, spotted onto nitro-cellulose filters placed onto LB agar supplemented as necessary with 0.2% arabinose for induction, and incubated for 5 h at 37˚C. Filters were suspended in 1 ml LB and vigorously vortexed to dislodge cells from the filters and interrupt mating. For liquid matings, 50 μl each of donor and recipient cultures grown as described above were mixed and incubated for 1 h at 37˚C, then vigorously vortexed. Mating mixes were serially diluted with LB, and serial dilutions were plated on LB agar containing antibiotics selective for transconjugants (Tcs), recipients, and donors. The frequency of DNA transfer is presented as the number of transconjugants per donor (Tcs/D). All matings were repeated at least three times in triplicate; a representative experiment is shown with replicate data points and the average transfer frequencies as bars along with standard deviations as error bars. For matings with donors bearing coresident pED208 and F plasmids, donor and recipient cells were grown overnight at 37˚C in presence of the appropriate antibiotics, diluted 1:50 in fresh antibiotic-free LB media, and incubated with shaking for 2 h. 75 μl of donor and recipient cell cultures were mixed in 96 well plates and incubated for 3 h at 37˚C, and the mating mix was serially diluted with LB and plated on LB agar containing antibiotics selective for transconjugants acquiring each of the two plasmids and donors. Source data for all conjugation experiments are presented in **S5 Table**.

### Phage infection assays

Sensitivity of plasmid-carrying host cells to the M13KO7 and MS2 bacteriophages was assessed as previously described [25]. Briefly, strains carrying F or pED208 plasmids were grown overnight in the presence of the appropriate antibiotics, diluted 1:50 in fresh antibiotic-free LB medium, and incubated for 3.0 h. A 1 ml aliquot of cells was incubated with 1 μl M13KO7 ($10^{11}$ plaque-forming units per ml, pfu/ml) (NEB) for 10 min on ice to allow attachment, and then at 37˚C for 10 min to allow infection. Cells were washed once with fresh LB and incubated for an additional 30 min at 37˚C. Cells were serially diluted and plated onto LB agar containing antibiotics selective for Kan$^r$ transductants and for total colony-forming units (CFUs). M13KO7 sensitivity was calculated as the number of Kan$^r$ transductants/total CFUs. M13KO7 phage infection experiments were performed at least three times in triplicate, with results presented for a single representative experiment showing replicate data points, average transfer frequencies as horizontal bars, and standard deviations as error bars. For M13KO7 infections, quantitative data are presented in **S3 Fig**, and presented in the manuscript as '+' (sensitive, defined as $>10^{-4}$ Kan$^r$ colonies/total CFUs), '-' (resistant, defined as $<10^{-6}$ Kan$^r$/total CFUs), or '+$^p$' (partially sensitive, $10^{-4-6}$ Kan$^r$/total CFUs). S3 Fig source data are presented in **S5 Table**. For determinations of MS2 phage sensitivity, a 500 μl aliquot of cells subcultured as described above was mixed with 5 ml LB soft (0.75%) agar. A 5 μl aliquot of MS2 (~$10^{11}$ pfu/ml) (kindly provided by L. Zeng, Texas A&M) was spotted and the plate was incubated overnight and examined for plaque formation. Results are presented as '+' (sensitive, clear plaques), '+$^p$' (partially sensitive as evidenced by turbid plaques) and '-' (resistant, no plaque formation) [25].

### Detection of Tra proteins by immunostaining

Overnight cell cultures (5 ml) were diluted 1:100 in fresh LB and incubated for 1.5 h at 37˚C. Sodium salicylate (1 μM final concentration; VWR) was added for induction of *traD*, *traJ*, or *traB* variants, and cells were incubated for 1 h at 37˚C. Cultures (1 ml) were centrifuged at 5,000 *x g* for 5 min to pellet cells and resuspended in 50 μl of physiologically buffered saline (PBS). An equivalent volume of 2x Laemmli buffer was added, and samples were boiled for 5 min prior to electrophoresis through SDS-12.5% polyacrylamide (30:0.8 acrylamide/bis-

acrylamide) gels. For detection of TraA pilins encoded by pED208 or F, overnight cell cultures (5 ml) were diluted 1:100 in fresh LB, incubated at 37˚C for 2 h, and 1 ml aliquots of cell cultures were harvested by centrifugation at 5,000 $x\ g$ for 15 min. Cell pellets were resuspended in 50 μl of PBS and 50 μl of 2x Laemmli's sample buffer and boiled for 5 min. Samples were electrophoresed through SDS-16.5% polyacrylamide (30:0.8 acrylamide/bis-acrylamide) gels. Electrophoresed material was transferred to nitrocellulose membranes and blots were developed with primary antibodies against the Strep-tag (Genscript) for detection of Strep-tagged TraD, TraJ, or TraB variants, or with anti-TraA polyclonal or JEL-92 monoclonal antibodies specific for $TraA_{ED}$ or $TraA_F$, respectively (kindly provided by L. Frost). As a loading control, all blots were also developed with antibodies against *E. coli* RNA polymerase β subunit (RNP). Blots were developed with horseradish peroxidase (HRP)-conjugated secondary antibodies for detection of immunostained proteins by chemiluminescence. The entire protocol was repeated at least three times, and representative immunoblots are presented.

### Detection of ED208 pili by AF488-maleimide labeling and F pili by decoration with MS2-GFP

ED208 pili were fluorescently labeled as previously described [25]. Briefly, host cells carrying pED208 variants and pKKF005 (expresses *traAcys*) were grown overnight, diluted 1:50 in fresh antibiotic-free LB medium, and incubated for 1.0 h at 37˚C. As appropriate, cells were induced for *traAcys* expression with 1mM sodium salicylate (final concentration), and incubated for 2.5 h at 37˚C [25]. AlexaFluor 488 $C_5$ Maleimide (AF488-mal) (Thermo Fisher Scientific) was added to a 100 μl aliquot of the cell culture at a final concentration of 25 μg/ml followed by incubation for 30 min on ice. Cells were washed once in physiologically-buffered saline (PBS), and resuspend in 100 μl PBS. Labeled cells (5 μl) were placed on a 1% PBS agarose pad for imaging with Nikon A1 confocal microscope with a Plan Apo 100x objective lens (Oil Immersion), and NIS-Elements AR software with FITC and DIC filters). F pili were visualized by adsorption of fluorescent MS2-GFP [25,47]. Host cells carrying F variants were grown overnight, diluted 1:50 in fresh antibiotic-free LB medium, and incubated for 3.5 h at 37˚C. A 5 μl aliquot of cells was mixed with MS2-GFP ($10^{10}$ pfu's/ml final concentration) and incubated for 30 min on ice. Labeled cells were placed on a 1% PBS agarose pad for imaging with a Nikon A1 confocal microscope. At least 250 cells per strain were examined to quantitate the percentage of cells in a population with visible ED208 or F pili.

### Pairwise structure alignment

The TraA pilin and $TraB_{ED}$ structures were obtained from the RCSB Protein Data Bank web server (wwpdb.org) [81]. Structures of the Tra and Trb homologs encoded by F and pED208 were predicted using the AlphaFold Protein Structure Database (https://alphafold.ebi.ac.uk/) [34,82]. Predictions of the $TraJ_{KM}$ structure were generated using standard settings and databases for ColabFold (https://colab.research.google.com/github/sokrypton/ColabFold/blob/main/AlphaFold2.ipynb#scrollTo = kOblAo-xetgx) [83]. The highest-ranking TraJ protein structure was depicted. Comparative model constructions of the Tra and Trb homologs were performed with the Pairwise Structure Alignment tool (https://www.rcsb.org/alignment) on the RCSB Protein Data Bank web server with the parameter set to jFATCAT. The generated pairwise structure alignments were modified using ChimeraX-1.5 [84].

### Bioinformatics analyses

We used the COMPASS database [76] to identify the numbers of F plasmids carried by bacterial strains identified as having at least one F-like plasmid. We identified a total of 908 bacterial

strains as carrying at least one plasmid categorized as IncF. The fractions of strains harboring one or more F-like plasmids was presented in a pie chart format. We generated a presence/absence profile for the *tra/trb* genes using the IncF plasmid dataset (5,664 sequenced plasmids) obtained from PLSDB (v. 2021_06_23_v2) [79]. We carried out tblastn searches against a nucleotide database constructed from the IncF plasmid sequences dataset using the following Tra/Trb proteins as queries: TraA, TraL, TraE, TraK, TraB, TraV, TraC, TraF, TraW, TraU, TrbC, TraN, TraQ, TrbB, TrbI, TraH, TraG and TraD carried by pOX38 (MF370216.1), pED208 (AF411480.1), pKpQIL-UK (KY798507.1), and pOZ172 (CP016763.1). We selected these plasmids as queries because ~90% of the F plasmids analyzed here were from one of four enterobacterial species and the plasmid queries also were originally identified in these species (F, *E. coli*; pED208, *Salmonella spp.*; pKpQIL *Klebsiella spp.*; pOZ172, *Citrobacter spp.*). By default, hits greater than an e-value of $10^{-3}$ were considered as positive hits. When there was a hit using at least one query, the gene was defined as present. Pie charts are presented for the fractions of F plasmids with complete sets of *tra/trb* genes vs those missing 1 or more, and for the fractions of plasmids missing specific *tra/trb* genes among the collection of F plasmids shown to lack 1 *tra/trb* gene. All source data for these bioinformatics analyses are presented in S3 Table.

## Supporting information

**S1 Fig. Alignments of Tra/Trb protein homologs.** Panel A: Alignment of the F and pED208 *tra/trb* regions. Blue: VirB/VirD4-like scaffold subunits that are conserved among T4SSs. Red: F-specific proteins that contribute to assembly or function of F-like T4SSs. Yellow: F-specific proteins involved in surface exclusion. Green: Nonconserved proteins. Lines connect genes encoding protein homologs; numbers correspond to the percent amino acid identities across the lengths of the homologs. Accession numbers: pED208 (NCBI Bioproject PRJNA871772); F (NZ_MF370216). By comparison, most F/pED208 VirB/VirD4 subunits exhibit ~18–20% sequence identities with their counterparts in the 'minimized' pKM101-encoded system; outliers include the VirB6-like $TraG_F/TraD_{KM}$ (8% identical) and VirB7-like $TraV_F/TraN_{KM}$ (~6% identical) homologs. **Panels B—F (following pages):** Similarities among the Tra/Trb homologs at the primary sequence and structural levels, grouped as indicated (orange highlights). Upper: Sequence alignments of the Tra/Trb proteins indicated, generated with Multalin [85]. Known domains and percent identities are shown for regions indicated. Red lines denote regions of sequence divergence, one or more of which might confer the distinct phenotypes accompanying subunit swapping described in the text. Lower: Structures of the Tra/Trb homologs predicted from the AlphaFold Protein Structure Database [82]. The solved TraA and $TraB_{ED}$ structures were from RCSB PDB. The structure of pKM101-encoded TraJ was predicted by ColabFold [83]. Structures of the homologs are presented separately and superimposed. For TraB, only the β-barrel and AP domains comprising the OMCC and AP channel are shown. Percent identities of the structurally aligned sequences, Root Mean Square Deviation (RMSD), TM-score, and number of equivalent residues relative to the entire sequence lengths were derived from the RCSB PDB Pairwise Structure Alignment website (https://www.rcsb.org/alignment).
(PDF)

**S2 Fig. Functionality of p*oriT* plasmids and *traD* variants. A)** Transfer of p*oriT_{ED}* and p*oriT_F* plasmids by donor cells carrying pED208 (blue bars) or F (green bars). **B)** Functionality of Strep-TraD in donor cells with pED208 (blue bars) or F (green bars) variants. Donor cells carried pED208 or F or the isogenic Δ*traD* mutant plasmids without or with plasmids expressing the corresponding *traD* or *strep-traD* genes. **Panels A & B)** Transfer frequencies are presented

as transconjugants per donor (Tcs/D). All matings were repeated at least three times in triplicate; a representative experiment is shown with replicate data points and the average transfer frequencies as horizontal bars along with standard deviations as error bars. **Panels C & D)** Host cells carrying pED208Δ*traD* were assayed for production of the strep-tagged $TraD_{ED}$, $TraD_F$ or $TraJ_{KM}$ variants shown. Total cellular proteins normalized on a per cell equivalent basis were subjected to SDS-PAGE and immunostaining of western blots with α-strep antibodies for detection of the T4CP variants or α-RNP antibodies against *E. coli* RNA polymerase β-subunit as a loading control.
(PDF)

**S3 Fig. Quantitation of strain sensitivities to M13KO7 (source data).** M13KO7 phage sensitivity is shown for host cells carrying F or pED208 or gene deletion variants without or with a plasmid expressing the corresponding genes from F or pED208. Phage sensitivity is reported as the number of kanamycin-resistant ($Kan^r$) transductants per total colony-forming units (CFUs). Panels: **A.** *traD* T4CPs; **B.** IMC subunits; **C.** OMCC subunits; **D.** F-specific components; **E.** *traA* pilins. All infection assays were repeated at least three times in triplicate; a representative experiment is shown with replicate data points and the average transfer frequencies as vertical bars along with standard deviations as error bars. Data in the manuscript figures are presented as '+' (sensitive, defined as $>10^{-4}$ $Kan^r$ colonies/total CFUs), '-' (resistant, defined as $<10^{-6}$ $Kan^r$/total CFUs), or '$+^p$' (partially sensitive, $10^{-4-6}$ $Kan^r$/total CFUs). Source data appear in S5 Table.
(PDF)

**S4 Fig. Transfer of donors harboring Tra+,Pil- "uncoupling" mutations in liquid matings (A) and production of TraB variants and $TraA_{ED}$ pilin in different mutant backgrounds (B-D). A)** pED208 or F transfer frequencies by donors bearing "uncoupling" mutations in 1 h liquid matings; matings were repeated at least three times in triplicate and a representative experiment is shown with the average transfer frequencies as horizontal bars along with standard deviations as error bars. Colored bars denote source of plasmid or expressed gene of interest (blue, pED208; green, F; chimera, blue/green checkered); for comparison, gray bars represent transfer frequencies for the same strains in solid-surface matings as reported in the manuscript. **B)** Schematics of TraB chimeras and deletion mutants. Chimeras consist of domains from $TraB_{ED}$ (blue) and $TraB_F$ (green). Host cells carrying pED208Δ*traB* and complementing plasmids were assayed for production of the strep-tagged TraB variants shown. Total cellular proteins normalized on a per cell equivalent basis were subjected to SDS-PAGE and immunostaining of western blots with α-strep antibodies for detection of the TraB variants or α-RNP antibodies against *E. coli* RNP β-subunit as a loading control. **C)** Production of $TraA_{ED}$ pilin in strains carrying pED208 or mutant plasmids deleted of the *tra/trb* genes shown; Lane 2: *E. coli* MC4100 expressing $traA_{ED}$ from pKKF004 in the absence of pED208. **D)** Schematic of the pED208 *tra/trb* region with the ΔOMCC and ΔF-specific deletion mutations highlighted; production of $TraA_{ED}$ pilin in strains with the pED208ΔOMCC and pED208ΔF-specific mutant plasmids. **Panels C & D)** $TraA_{ED}$ pilin was detected by immunostaining with α-$TraA_{ED}$ polyclonal antibodies (upper and lower bands are crossreactive species) and RNP β-subunit with α-RNP antibodies.
(PDF)

**S1 Table. Strains and plasmids used in this study.**
(PDF)

**S2 Table. Oligonucleotides used in this study.**
(PDF)

**S3 Table. Source data for bioinformatics analyses presented in Fig 7A.**
(XLSX)

**S4 Table. Conjugation proficiencies and M13KO7 susceptibilities of strains harboring co-resident F and pED208 mutant plasmids presented in Fig 7B.**
(PDF)

**S5 Table. Source data for all conjugation and M13KO7 infection assays.**
(XLSX)

## Acknowledgments

We thank Laura Frost for providing α-TraA antibodies and strains carrying pOX38, pOX38-Km-traQ238, or pED208. We thank Lanying Zeng for helpful discussions and for purified MS2 phage. We thank Bill Margolin for the gift of pKG116.

## Author Contributions

**Conceptualization:** Kouhei Kishida, Yang Grace Li, Natsumi Ogawa-Kishida, Peter J. Christie.

**Data curation:** Peter J. Christie.

**Formal analysis:** Kouhei Kishida, Yang Grace Li, Peter J. Christie.

**Funding acquisition:** Peter J. Christie.

**Investigation:** Kouhei Kishida, Yang Grace Li, Natsumi Ogawa-Kishida, Pratick Khara, Abu Amar M. Al Mamun, Rachel E. Bosserman, Peter J. Christie.

**Methodology:** Kouhei Kishida, Yang Grace Li, Natsumi Ogawa-Kishida, Pratick Khara, Abu Amar M. Al Mamun, Rachel E. Bosserman, Peter J. Christie.

**Project administration:** Kouhei Kishida, Peter J. Christie.

**Resources:** Kouhei Kishida, Peter J. Christie.

**Supervision:** Peter J. Christie.

**Validation:** Kouhei Kishida, Yang Grace Li, Natsumi Ogawa-Kishida, Pratick Khara, Abu Amar M. Al Mamun, Rachel E. Bosserman.

**Visualization:** Kouhei Kishida, Yang Grace Li, Natsumi Ogawa-Kishida, Peter J. Christie.

**Writing – original draft:** Kouhei Kishida, Yang Grace Li, Peter J. Christie.

**Writing – review & editing:** Kouhei Kishida, Yang Grace Li, Natsumi Ogawa-Kishida, Rachel E. Bosserman, Peter J. Christie.

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
