## [Decision Letter · Decision Letter 0]

11 Jan 2024

Dear Dr Christie,

Thank you very much for submitting your Research Article entitled 'Chimeric systems composed of swapped Tra subunits between distantly-related F plasmids reveal striking plasticity among type IV secretion machines' to PLOS Genetics.

The manuscript was fully evaluated at the editorial level and by independent peer reviewers. The reviewers appreciated the attention to an important topic but identified some concerns that we ask you address in a revised manuscript.

We therefore ask you to modify the manuscript according to the review recommendations. Your revisions should address the specific points made by each reviewer.

Yours sincerely,

Ankur B. Dalia

Academic Editor

PLOS Genetics

Sean Crosson

Section Editor

PLOS Genetics

Editor's comments: 

As you will see in the reviews below, all three reviewers are enthusiastic about the initial submission. But all have highlighted some minor points for you to consider, which should be addressed by changes to the text, additional analysis of existing data, and/or by performing additional experiments as indicated.

Reviewer's Responses to Questions

**Comments to the Authors:**

Reviewer #1: Bacterial conjugation is a paramount research topic due to its significant contribution to the dissemination of antibiotic resistance in bacteria. Autonomous conjugation plasmids encode the machinery enabling their transmission to other bacterial cells, including a conjugative type IV secretion system (T4SS), strictly essential to transport the DNA through the membranes of the donor and recipient cell engaged in the mating process. T4SSs from different conjugation plasmids are composed of various conserved subunits, yet they also exhibit diversity in composition and structure. T4SS gene systems encode a type IV coupling protein (T4CP), an inner membrane complex (IMC), an outer membrane core complex (OMCC), and factors involved in the production of the conjugative pilus.

In the manuscript "Chimeric systems composed of swapped Tra subunits between distantly-related F plasmids reveal striking plasticity among type IV secretion machines," Kishida and co-workers report the results of an ambitious study aiming at clarifying how T4SSs recruit specific substrates, elaborate functional channels, and in some cases build the conjugative pili. To do so, they

implemented a well-designed genetic dissection involving systematic deletion/complementation and subunit swapping, and they assessed the functionality of the resulting chimeric machines. They proceed by swapping individual Tra/Trb subunits between T4SSs encoded by the phylogenetically distantly related F-like plasmids, F and pED208. The chimeric systems are examined in terms of the ability to produce functional conjugation machinery as addressed by the efficiency of plasmid transfer) and ability to produce the conjugative pilus as reported by fluorescence microscopy imaging and sensitivity to M13 and MS2 phages, which adsorb the tip and to the side of the conjugative pilus, respectively. This study produces a wealth of data that helps understand the plasticity and cross-functionality of these two F-encoded T4SSs.

The main insights and conclusions that the authors put forward are - The identification of regions of intrinsic flexibility among the encoded T4SSs - The relaxed specificity between TraD and its substrate, and with the T4SS - the identification of which TraD domains mediates the recruitment of cognate or heterologous DNA substrates. The C15 motifs appear to be the most critical for recruiting cognate F-like plasmids. The transfer was less but still stimulated by the CTD�C15, and even less by the TraD�CTD - The unanticipated observation of DNA transfer in the absence of detectable pilus in some strains producing chimeric systems. On this last point, this reviewer thinks that the lack of detectable pili using microscopy and sensitivity to M13 infection does not strictly exclude the possibility that some pili might exist in a form, location, or configuration that would not be accessible to M13 phage or the maleimide conjugates. Therefore, some of the interpretations and conclusions might be moderated accordingly (see comments below) - Strains containing "incomplete" F-like and F plasmids (lacking T4SS components required for their autonomous transfer) do elaborate functional chimeric T4SSs enabling plasmid transfer. This elegant study leads to the conclusion that combinations, interactions, and complementarity between T4SS components from different plasmids likely facilitate the propagation of plasmids that are not self-transmissible per se.

Overall, the experimental design and results are sound and of high quality. Conclusions are well supported by the data (though potentially overstated regarding the absence of detectable pili). In summary, this work constitutes a significant addition to the field of conjugation biology and the understanding of the plasticity of type IV secretion machines.

Comments:

1- Figure 1Bi and ii: This might be due to a misunderstanding of the figure, but this reviewer feels that the transfer value shown for TraDED complementation in panel Bi should be the same as for TraDED complementation in panel Bii (as is the case for panel Ci and Cii).

Also, have the authors tryied to produce the traDEDC15F in the pED208del(traD) and vise versa, i.e., the traDFC15ED in the pFdel(traD)? Would that be an interesting addition to the data set by producing firther insights regardnig the role of the C15?

An additional minor remark is that the individual data points are hardly visible on the blue bars. Though this requires the editing of most figures, the authors could consider using a lighter blue.

2- L191-192: "Interestingly, donors harboring pED208deltraD and producing TraJKM supported transfer of pED208deltraD, albeit at a low frequency of ~10^-7 Tc's/D (Fig. 1D). We appended TraDED's CTD to TraJKM, which elevated pED208 transfer by ~10^1.5-fold.". This reviewer agrees that TraJKM enables low transfer frequency of the pED208�traD plasmid; however, the ~101.5-fold increase with the TraJKMCTDED is less convincing due to the low values and respective standard deviations. A statistical test might be required to ascertain the significance of such subtle differences and strengthen the interpretation.

3- L195-196: The statement "TraJKM efficiently mediates the transfer of poriTKM, a plasmid harboring the pKM101oriT sequence, through the T4SSKM" is supported by reference 37. It would be helpful to the reader to indicate the frequency achieved in this context (TC's/D = 10^0 if I am correct?). In the absence of TraD, the frequency drops to 10^-4. The following statement, L197-198, "…showing that TraJKM functionally interacts with both T4SSs…" might convey that TraJKM interacts equally with both T4SSs, which does not appear to be the case. Therefore, the authors might consider restating this conclusion to notify the difference.

4- L200-202: "TraD's NTD/NBD thus recruits a completely heterologous substrate, whereas the C15 motif blocks this recruitment. Finally, strains co-producing native TraDED or TraDF and TraJKM failed to transfer poriTKM, suggesting that the TraD T4CPs outcompete TraJKM for engagement with the cognate T4SS channels". Do the authors have any interpretation regarding the fact that poriTKM transfer is observed in the presence of TraJKM and TraDdelCTDED but not in the presence of TraJKM and TraDdelCTDF? Does that teach us anything about the differences between F and pED208 systems?

5- Figure 2B: This reviewer understands there might be a mistake in the annotation of the bottom six lines in panel B. If this reviewer is correct, these lines should indicate deltraCF and not ED; traCF and not ED; traCED and not FdeltraGF and not ED; traGF and not ED; traGED and not F.

6- L260-263: "Strains of interest were engineered to produce Cys-derivatized TraAED pilins (TraA.C116), shown previously to support pED208 transfer at WT levels [25]. The Cys118 residues are surface exposed on ED208 pili and accessible to labeling with fluorescent maleimide conjugates, e.g., FM488-mal. We and others also have deployed fluorescently-tagged ssRNA phages such as MS2 to decorate F pili [25, 46, 48]." For the sake of exhaustive reference to previous relevant works, the authors might consider citing the recent PMID:37963249 article, where functional fluorescent labeling of the pilus using maleimide conjugates has been reported for Cys-derivatized TraAED and TraAF pilins.

7- The addition of microscopy imaging to visualize the pili is a nice and informative addition to the phage infection test. This strengthens the unexpected observation that some chimeric systems can mediate DNA transfer despite the absence of apparent pili at the surface of the cells, as in the case of strains carrying pED208deltraK or FdeltraK. To this reviewer's knowledge and understanding, this is a novel and important observation that compels us to revise some aspects of our conjugation model. At least, it raises one immediate question. Since all conjugation tests have been performed on solid surfaces, it becomes tempting to assess if those cells can perform conjugation in liquid conditions, where the requirement for pili functions differs. Have the authors performed conjugation tests in liquid conditions? This would undoubtedly deepen the interpretation and impact of these results by providing further clues as to how conjugation might occur in the absence of detectable pili.

8- L265-267: The statement "The deltraK mutant hosts also were resistant to infection by M13K07, further showing that the deltraK machines do not even produce "vestigial" pili with only their tips exposed on the cell surface". The authors may or may not be right in their interpretation. In any case, at this stage, this reviewer feels that the word "showing" is excessive since one cannot exclude the possibility that a vestigial pilus would remain located within the T4SS channel without necessarily exposing any TraA pilin that would be accessible to the M13K107 phage (or to the maleimide conjugate for that matter). Therefore, the authors might consider tempering this interpretation.

Along the same line, this reviewer feels that the paragraph's conclusion, "These findings firmly establish that F-like pili are completely dispensable for efficient DNA transfer through F-like channels." should be made more cautious. Indeed, the results show that DNA transfer can occur in the absence of detectable pili (using microscopy and phage infection), but do not strictly exclude the potential existence of pili types/forms that would not be detected with those two tests. It would also be helpful to have a section explaining how the authors reconcile their finding that detectable pili are not required for transfer with the essentiality of TraA for DNA transfer in all tested genetic contexts (in this work and the literature).

9- The results reported in Figure 7 are pivotal to the work. In particular, panel 7B presents phage sensitivity and plasmid transfer efficiency of strains carrying co-resident pED208 or F plasmid with individual deletion of tra/trb genes. The results recapitulate and generalize the interpretations reported at the beginning of the article, thus showing the soundness of the entire work. However, this reviewer finds a set of results challenging to integrate and understand regarding F�traA). Since we have learned from Figure 6 that FdeltraA is not complemented by TraAED, how can we explain that FdeltraA conjugation deficiency is suppressed by the single deletion of deltraDED from the pED208 plasmid? Does this mean that FdeltraA plasmid transfer is mediated by TraDF through a T4SSF that produces a pilus with TraAED pilin? Can the author clarify this point and confirm or correct this interpretation?

10- The labels Fig. 5, 6, and 7 are missing at the bottom right corner of the corresponding figures.

11- The data of M13 infection tests are presented in the supplementary Figure S3 (except those reported in Figure 4) , however, none of the raw results for MS2 phage are presented in the manuscript. Can the authors comment on this point?

Reviewer #2: The manuscript by Kishida et al. investigates conjugation systems encoded by two distantly related IncF plasmids, pED208 and F, by systematically deleting components or generating chimeric systems with swapped Tra subunits. The experiments reveal domains of the TraD coupling proteins that contribute to recruitment of cognate or heterologous DNA substrates. Many of the chimeric T4SSs retained the capacity to transfer DNA but exhibited defects in pilus production. Overall, the manuscript is well-written and provides important new insights.

Comments:

1. Lines 82-89 describe structural features of the T4SS encoded by the conjugative plasmid R388, and line 93 describes a previous structural analysis of the OMCC encoded by the IncFV plasmid pED208. It would be helpful to clarify in the Introduction that the R388-encoded T4SS is classified as a prototype or minimized system, whereas the pED208-encoded system is classified as an expanded system.

2. Line 93 cites a previous structural analysis of the OMCC encoded by pED208 (reference 15, Liu et al. 2022). At least two additional papers have analyzed structural features of the T4SS encoded by pED208 (Amin et al., 2022, reference 14; and a cryo-ET study by Hu et al., PNAS 2019). These papers should also be cited on page 4. It would be helpful to comment in the Introduction whether structural analyses have previously been undertaken for both of the T4SSs analyzed in the current study (pED208 and F) or only the T4SS encoded by pED208.

3. Figure S1 provides valuable information comparing genetic features of the two conjugation systems analyzed in the current study. Fig. S1A illustrates that the Tra/Trb genes in these systems exhibit high levels of sequence divergence. To further orient readers, it would be helpful to briefly comment on the corresponding levels of sequence divergence when comparing components of prototype/minimized T4SSs (such as the R388-encoded system) with components of the conjugation systems analyzed in the current study. For example, are components of the two conjugation systems analyzed in the current study more closely related to each other than to components of prototype/minimized T4SSs such as the R388-encoded system?

4. Figure 1A shows components of chimeric systems in solid colors (green or blue) corresponding to their origin, whereas Figure 2 and several other figures show substituted components in blue/green checkered patterns. Suggest adding a brief description to the figure legends to help readers understand the difference between the solid and checkered color schemes.

5. Fig. 6C shows one band labeled as TraA-ED, as well as two other bands. Are the two unlabeled bands proteins that cross-react with the antiserum? Consider commenting on this in the figure legend.

Reviewer #3: This exceptional manuscript by Kishida et al. describes the generation and analysis of chimeric conjugative type IV secretion systems (T4SS) used to decipher the role of multiple components in apparatus biogenesis and interbacterial DNA translocation. Using interchangeable pED208 and F pilus constituents, the authors thoroughly investigated the mechanistic details underscoring machinery assembly, DNA substrate recruitment, secretion channel integrity, and conjugative pilus biogenesis. This work exploits well-controlled and extensive mutagenesis studies and systematic functional analyses to dissect protein architectures and domains that govern substrate recruitment and translocation channel assembly among artificial and endogenous T4SS chimeras. Notably, the authors convincingly demonstrate subunit interchangeability among isolates harboring non-selftransmissible co-resident T4SS variants, highlighting the clinical and evolutionary importance of these findings within the context of functional diversification and apparatus structural flexibility. Overall, the experimental approaches are rigorous, the manuscript is well written and logically presented, and the findings represent an exciting and novel advance in the T4SS field. In the interest of strengthening the conclusions, the authors should address the following minor comments:

1. In line 162, the authors speculate that truncation of TraD (lacking the C-terminal discrimination motif) attenuates translocation of pED208 and F through the cognate T4SS machineries due to loss of TraD-TraM interactions. Can this hypothesis be tested using bacterial two-hybrid or co-immunopurification approaches?

2. In experiments designed to analyze the contribution of TraB AP on DNA conjugation (lines 321 – 327), several pilus uncoupling mutations were identified that enabled substrate transfer in the absence of vestigial or extracellular pilus assembly. In addition to testing AP variants in which residues are substituted with glycine (e.g., TraBAPL5GED), the authors should analyze whether increased or decreased AP length or loop rigidity influences substrate transfer or pilus biogenesis. While the AP is required for conjugation (Fig. 4), does the AP length or local flexibility alter conjugation frequencies, translocation channel formation, or pilus assembly?

3. In lines 535 – 536, the authors propose that co-resident TraV or TraK are more readily incorporated into heterologous T4SS machineries after the subunits first adopt the correct folded structures within the cognate apparatus. In this scenario, do the authors suggest that OMC subunits disassociate/disassemble and re-assemble into adjacent or nearby heterologous systems? Could disassembled subunits remain in the correctly folded conformation in the periplasm/outer membrane and stochastically incorporate into nascent heterologous subcomplexes/subassemblies during apparatus biogenesis? This speculative model should be more completely elaborated.

4. In light of two very recent papers describing ssDNA transfer through the F pilus (PMID: 37963249) and pED208 (PMID: 38051116), the results of this study represent a significant advance supporting a model of high efficiency DNA conjugation in the absence of observable pilus biogenesis. Given the results of the current study, can the authors provide additional comment/speculation regarding the role of pili versus proximal mating junctions in substrate delivery? Perhaps the major role of pilins is to stabilize the secretion channel during substrate transfer (while extracellular pili mediate low efficiency DNA translocation)?

**Have all data underlying the figures and results presented in the manuscript been provided?**

Reviewer #1: Yes

Reviewer #2: Yes

Reviewer #3: Yes

PLOS authors have the option to publish the peer review history of their article (what does this mean?). If published, this will include your full peer review and any attached files.

Reviewer #1: **Yes: **Christian LESTERLIN

Reviewer #2: No

Reviewer #3: No

---

## [Editor Report · Decision Letter 1]

20 Feb 2024

Dear Dr Christie,

We are pleased to inform you that your manuscript entitled "Chimeric systems composed of swapped Tra subunits between distantly-related F plasmids reveal striking plasticity among type IV secretion machines" has been editorially accepted for publication in PLOS Genetics. Congratulations!

Yours sincerely,

Ankur B. Dalia

Academic Editor

PLOS Genetics

Sean Crosson

Section Editor

PLOS Genetics

Comments from the reviewers (if applicable):

**Data Deposition**

http://datadryad.org/submit?journalID=pgenetics&manu=PGENETICS-D-23-01340R1

**Press Queries**

---

## [Editor Report · Acceptance letter]

28 Feb 2024

PGENETICS-D-23-01340R1 

Chimeric systems composed of swapped Tra subunits between distantly-related F plasmids reveal striking plasticity among type IV secretion machines 

Dear Dr Christie, 

We are pleased to inform you that your manuscript entitled "Chimeric systems composed of swapped Tra subunits between distantly-related F plasmids reveal striking plasticity among type IV secretion machines" has been formally accepted for publication in PLOS Genetics! Your manuscript is now with our production department and you will be notified of the publication date in due course.

With kind regards,

Judit Kozma

PLOS Genetics

On behalf of:
